# REPROCESSED 2-D AIRGUN SEISMIC REFLECTION DATA SALTFLU (SALT DEFORMATION AND SUB-SALT FLUID CIRCULATION IN THE ALGERO-BALEARIC ABYSSAL PLAIN) IN THE BALEARIC PROMONTORY AND THE ALGERIAN BASIN

Simon Blondel[1,2], Jonathan Ford[1], Aaron Lockwood[3], Anna Del Ben[2], Angelo Camerlenghi[1]

[1]National Institute of Oceanography and Applied Geophysics – OGS, Trieste, Italy
[2]Department of Mathematics and Geosciences, University of Trieste, Trieste, Italy
[3]Shearwater GeoServices Inc, Gatwick, United Kingdom

*Correspondence to*: Simon Blondel (simon.blondel2@gmail.com)

**Abstract.** In an ever more challenging context for the acquisition of seismic data in the Mediterranean Sea, reprocessing to improve the quality of legacy data has become increasingly important. This work presents the newly reprocessed, open access dataset SALTFLU acquired in the Algerian basin by the National Institute of Oceanography and Applied Geophysics (OGS) in 2012. We apply a 'broadband' reprocessing strategy adapted for short-offset, deep water airgun reflection seismic data and assess if the reprocessed images provide new geological insights on the Mediterranean sub-surface. The workflow relies on an integrated approach combining geophysics and geological interpretation to iteratively build the velocity model. In this way we aim to tackle some of the challenges linked to imaging complex geological structures containing high velocity contrasts with 2-D, short-offset seismic data. We first broaden the bandwidth of the data through multi-domain de-noising, deghosting and a source designature using an operator derived from the seabed reflection. We then perform iterative migration velocity analysis, pre-stack time migration and multiple attenuation in the Radon domain to obtain time-migrated images. The initial velocity model is derived from the resulting time migration velocities, and geologically driven model updates are generated using a combination of travel-time tomography, seismic interpretation of the major salt horizons and velocity gradient flooding. The gradient flooding aims to reproduce the large scale first-order velocity variations, while the travel-time tomography aims to resolve the smaller second-order velocity variations. The results improve our deep geological knowledge of the under-explored Algerian basin down to the base salt and the pre-salt. Fluid indicators are imaged within the Plio-Quaternary of the Algerian basin, which we interpret as thermogenic or biogenic gas sourced from either the Messinian Upper Unit or from the pre-salt, migrating through a hydro-fractured salt. The reprocessed data image lateral and vertical seismic facies variation within the Messinian units that could shed new light on the tectono-stratigraphic processes acting during the Messinian Salinity Crisis. It also reveals numerous previously unresolved volcanic structures within the Formentera basin.

## 1. Introduction

The Mediterranean Salt Giant (MSG) is a vast and thick evaporitic unit that was deposited during the Messinian Salinity Crisis (MSC), an extreme environmental episode that occurred from ~5.97 to ~5.33 Ma (Hsü et al., 1973; Krijgsman et al., 1999; CIESM, 2008; Manzi et al., 2013). Since its discovery in 1970, and despite intense multi-disciplinary research, the MSG and the related MSC are still poorly understood and subject to many unresolved controversies (Camerlenghi and Aloisi, 2019). This is in large part due to a lack of data in the deep-water offshore domain, which represents more than 80% of the volume of MSC-related sediments, emphasizing the need for new and improved offshore data to better constrain the MSC (Roveri et al., 2014). Multi-channel seismic reflection profiling is the most common geophysical method applied to image the architecture of offshore basins and to prospect for potential

drilling sites. Acquiring new marine seismic data, however, is challenging due to high acquisition costs associated with marine operations. Numerous offshore seismic datasets have been acquired in the last decades in the Mediterranean Sea, providing countless 2-D profiles that could no longer be acquired today because they cover areas that are currently subject to restrictions related to obtaining exploration permits (Diviacco et al., 2015). Many of these datasets are currently poorly exploited due to lack of public data access or poor-quality seismic processing. Reprocessing legacy data, therefore, is a potential source of new geological information that could be extracted from these dormant datasets, directly contributing to a better understanding of the MSG and the MSC.

The SALTFLU ('Salt deformation and sub-salt fluid circulation') 2-D multi-channel seismic reflection dataset was acquired in June-July 2012 by the R/V OGS-Explora, Eurofleets Cruise No. E12 (acquisition parameters listed in Table 1). The survey was planned to study the influence of the MSG on pore fluid circulation during basin evolution since the post-Messinian. The legacy SALTFLU processing followed a 'narrowband' approach, without deghosting, using narrow bandpass filters coupled with a source designature based on statistical deconvolution and no zero-phasing of the target wavelet. The filtering eliminated much of the low frequency signal (below around 6 Hz), whilst the source designature boosted the high frequency noise and produced a mixed-phase wavelet that has inconsistent phase across the survey. The wavelet also contains strong residual energy (likely from the bubble pulse) that overprints and obscures the primary signal, particularly the shallow geology close to the water bottom (Jovanovich et al., 1983; Sheriff and Geldart, 1995; Yilmaz, 2001).

Common goals of reprocessing are to improve the data bandwidth, spatial resolution, signal-to-noise ratio, reflector continuity and, where relevant, seismic-borehole correlations (Sadhu et al., 2008; Lille et al., 2017). In this study we aim to better image the salt structures, particularly the base salt, previously interpreted from the legacy processing as a flat surface lying around 2.7 km below the seafloor, at a water depth of 3.2 km (Dal Cin et al., 2016). Due to the complex overburden geology and the short far-offset (3.1 km) with respect to the target depth, the main imaging challenges include accurately resolving velocity variations, eliminating multiples and improving the signal-to-noise ratio at depth. We confront these challenges by outlining three key stages that should be systematically included in modern processing flows for similar marine seismic datasets: i) 'broadband' processing, ii) multi-domain denoising and demultiple, and iii) geologically guided velocity model building using iterative pre-stack migration and travel-time tomography.

Compared to traditional 'narrowband' seismic processing, 'broadband' processing aims to improve the spectral accuracy by expanding the data bandwidth and restoring frequency content attenuated by the source- and receiver-side 'ghost' effect and by seismic absorption (Masoomzadeh et al., 2013; Lille et al., 2017). High frequencies are most strongly affected by seismic absorption, which preferentially attenuates the highest frequency parts of the spectrum (Futterman, 1962; Sams et al., 1997). Recovering this information improves the resolution and results in more accurate true amplitudes, improving the performance of other data processing steps such as velocity model estimation and migration, and quantitative interpretation such as amplitude variation with offset (AVO) analysis, impedance inversion, and attribute analysis (Chopra and Marfurt, 2007; Mavko et al., 2009; Amundsen and Zhou, 2013; Lille et al., 2017). Conversely, lack of low frequency signal commonly results in poor focusing in the deep part of the section, as low frequencies suffer less from scattering and absorption, so they penetrate deeper and display a better trace-to-trace moveout coherence, allowing us to build a more accurate velocity model (ten Kroode et al., 2013). In marine seismic data acquired with an airgun source, the source signature is a combination of a relatively broad impulsive signal (approximately a minimum-phase wavelet), periodic oscillations caused by the so-called 'bubble pulse' and an inverted polarity 'ghost' multiple, caused by the time-delayed reflection of the signal from the sea surface (Ziolkowski, 1970;

Sheriff and Geldart, 1995; Hegna and Parkes, 2011; Watson et al., 2019). The data bandwidth is principally widened by performing a source designature, whereby the primary airgun impulse and the bubble pulse are collapsed into a sharp, zero-phase wavelet (Sheriff and Geldart, 1995; Amundsen and Zhou, 2013; ten Kroode et al., 2013; Baldock et al., 2013; O'Driscoll et al., 2013). Deterministic deconvolution, where the operator is designed using an estimated source wavelet, can yield geological imaging superior to traditional statistical deconvolution methods, particularly for recovery of low frequencies and the preservation of amplitude information (Yilmaz, 2001; Sargent et al., 2011; Scholtz et al., 2015; Davison and Poole, 2015). Deghosting, instead, aims to deconvolve both the source- and receiver-side ghosts from the wavefield, further sharpening the wavelet and removing the frequency 'notches' associated with the ghost effect (e.g., Sargent et al., 2011; Chuan et al., 2014; Davison and Poole, 2015; Tyagi et al., 2016; Willis et al., 2018).

The quality of the bandwidth enhancement depends on the signal-to-noise ratio of the input data (Amundsen and Zhou, 2013). It is therefore essential to attenuate as much as possible the low frequency noise (e.g., reverberation from the direct arrival, 'swell' noise caused by pressure fluctuation near the sea-surface) beforehand, because any remaining low frequency noise not correlated with the source pulse may be artificially boosted by the source deconvolution and deghosting filters (Yilmaz and Baysal, 2015). Thanks to lower computational costs in recent decades, multi-channel filtering and analysis in transform domains has become routine for noise reduction (Schultz, 1985). For example, 'swell' noise can often be better attenuated by predictive filters in the frequency-space (F-X) domain than by a simple time domain low-cut filter that results in loss of the low frequency signal along with the attenuated noise (Liu and Goulty, 1999; Schonewille et al., 2008). Multiple attenuation is also better tackled by move-out discrimination techniques in the parabolic Radon domain rather than by traditional statistical deconvolution based methods, for example (Basak et al., 2012; Verschuur, 2013). A 'broadband' processing flow combined with an efficient multi-domain noise separation can improve the signal-to-noise ratio in the deep part of the section (in our case below the MSG), allowing considerable improvements in velocity model building (Chuan et al., 2014).

Seismic imaging restores the correct geometry of seismic reflectors and requires an accurate velocity model of the subsurface (Jones and Davison, 2014; Jones, 2015). The legacy SALTFLU data were imaged using a Kirchhoff pre-stack time migration. Time domain migrations are relatively robust to errors in the velocity model but are only well-suited to imaging geology containing weak lateral velocity variation (i.e., approximately 'layer cake' geology), as they do not properly account for ray path refraction. This can lead to degradation in image quality in time domain images of complex geology such as salt diapirs. Depth domain migrations, instead, can more accurately reproduce the ray paths of reflections in the subsurface, but the image quality is more sensitive to velocity errors (Sheriff and Geldart, 1995; Yilmaz, 2001; Jones and Davison, 2014). Migration velocities are generally estimated based on the flatness of reflections on common midpoint gathers after migration (Tsvankin and Thomsen, 1994; Jones, 2015). In depth domain, the process of picking reflectors is often automated and used as input to travel-time tomography. Iterative rounds of analysis of the residual curvature of reflectors on depth migrated gathers followed by travel-time tomography to calculate model updates (Jones, 2015).

In this study, we aim to showcase a 'broadband' reprocessing strategy designed to improve imaging of the MSG for the SALTFLU dataset. We demonstrate multi-domain de-noising, deghosting and a source designature using a seabed reflection derived operator. We then perform multiple attenuation and geologically driven iterative migration velocity analysis. Our results include pre-stack time and depth migrated images, which we compare to the legacy 'narrowband' processing. With these reprocessed images we highlight some new geological insights that these new results provide on the salt system of the Algerian basin, the seismic expression of the MSC and the basement structure of the study area.

## 2. Geological setting

The study area is located south of the Balearic Islands (Spain) with water depths between 1000 and 2800 m (Figure 1). The area covers the transition from the Formentera basin, on the Balearic Promontory, to the deep Algerian basin, which is marked by the steep NE-SW Emile-Baudot Escarpment (EBE), a NW-SE volcanic transform fault system (Acosta et al., 2001). The Algerian basin has previously been described as a Neogene back-arc oceanic basin that opened in response to the roll-back of the subduction of the Alpine–Maghrebian Tethys (Rehault et al., 1984; Verges and Sabat, 1999; Mauffret et al., 2004).

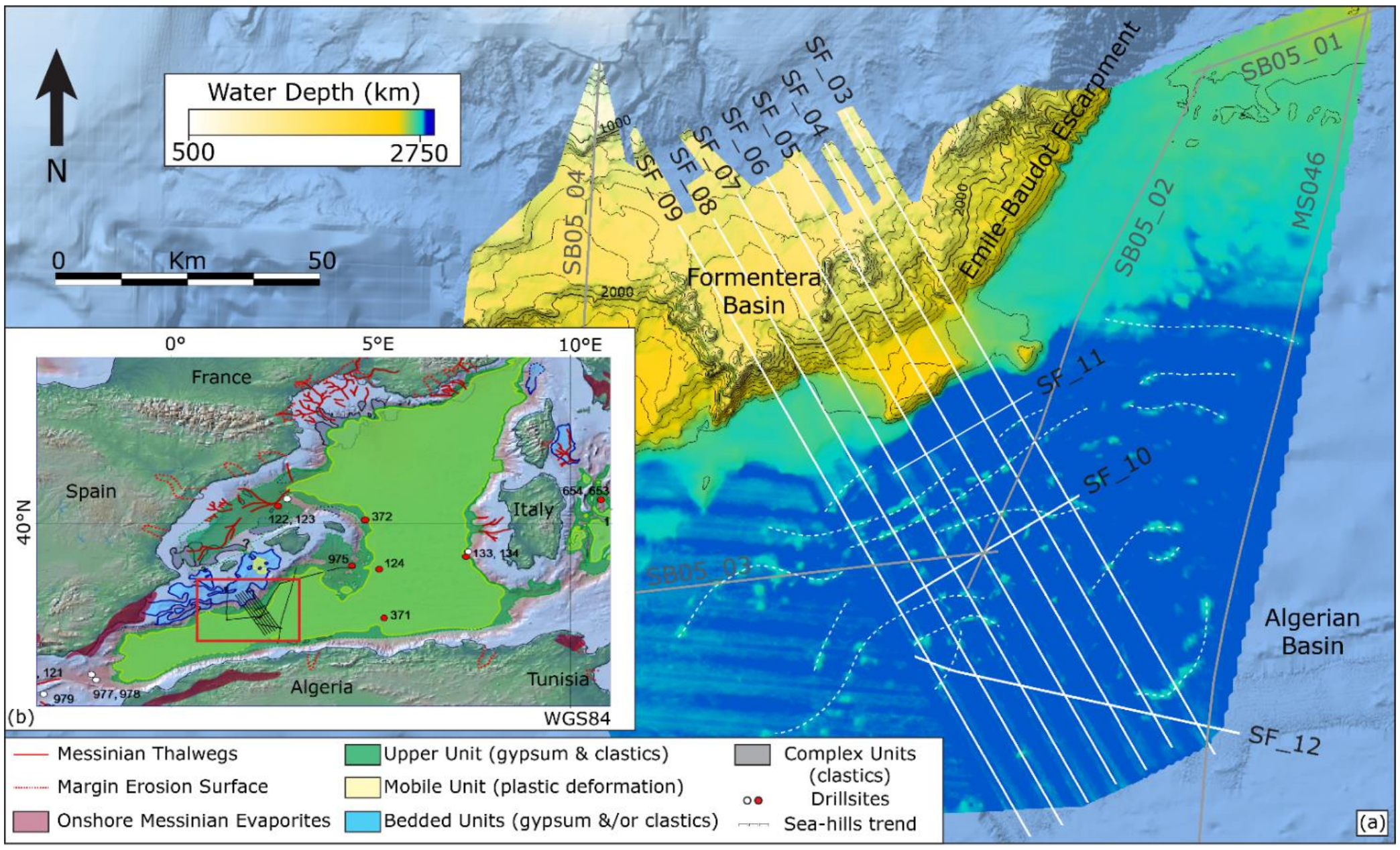

**Figure 1 Study area offshore Balearic Islands, Spain. Seismic lines acquired by the National Institute of Oceanography and Applied Geophysics (OGS) are superimposed on a bathymetric map and on the relief map of the Mediterranean with DSDP and ODP drillsites (in red the ones that sampled MSC evaporites) superimposed to the present-day spatial extent of the MSC marker (Lofi, 2018). Topographic highs on the sea-bottom illustrate the presence of piercing diapiric structures. SF refers to SALTFLU lines presented in this paper, SB05 refers to survey SBAL-DEEP 2005, MS046 to profile 46 of the Mediterranean survey.**

There are no boreholes within the study area to tie to the seismic profiles or to constrain the velocity model. The nearest borehole, Alger-1, is located 60 km to the south-east, in a perched basin, at only 100 m water depth, in a different geological setting (Burollet et al., 1978). The closest borehole located in the deepwater Algerian basin is the ODP Site 975, more than 200 km away from the line, that penetrated only the uppermost Messinian sediments (Comas et al., 1996).

Previous studies based their geological interpretation on the comparison with the analogous western Mediterranean margins, describing an Oligo-Miocene to Plio-Quaternary sediment cover, with the presence of thick (locally up to 2 km) Messinian evaporitic units both on the Balearic Promontory and in the deep offshore Algero-Balearic basin (Wardell et al., 2014; Driussi et al., 2015; Dal Cin et al., 2016; Camerlenghi et al., 2018; Raad et al., 2020). The Mobile salt Unit (MU) is identified in both the deep Algerian basin and the intermediate depth Formentera basin. In the Formentera basin, it separates the Bedded Unit 2 (BU2) from the Bedded Unit 3 (BU3; Raad et al., 2021). Laterally, when the salt pinches out, the BU2 turns into the Bedded Unit 1 (BU1) and/or the BU3. In the deep basin, the

MU is highly deformed, with complex and steep salt structures locally deforming the seafloor, and separates the Messinian Upper Unit (UU) from the pre-salt unit (Camerlenghi et al., 2009; Dal Cin et al., 2016; Camerlenghi et al., 2018).

## 3. Methods

Reprocessing of the SALTFLU dataset is based on the following major processing stages (Figure 2):

1. Noise attenuation
2. Bandwidth enhancement, including source designature and source- and receiver-side deghosting.
3. Multiple attenuation, pre- and post-migration.
4. Iterative pre-stack migration and velocity model building in time and depth domains.
5. Post-migration processing, including seismic attenuation compensation.

| Quality Check | Processing Step | Navigation Processing |
|---|---|---|
| | **Input Field Data** | **Import raw navigation** |
| **Cross-plot maps** | **Geometry merging** | ← **Edit Navigation**<br>Match FFID numbers to the corresponding shot numbers based on observer's logs |
| **Fold distribution, Stack** | **CMP Binning** | |
| **Output difference, Stack** | **1st pass Low Frequency Noise Attenuation on shot gathers**<br>Butterworth filter 2/30dB/Oct - 110/96dB/Oct | |
| **Time direct wave first break on near channel** | **Recording delay shift statics**<br>Shift statics of -40 ms | |
| **Visual checking** | **Resampling to 4 ms** | |
| **Output difference, Stack** | **FX Prediction Filtering (CRG)**<br>Run 1: 0-125Hz, 20 traces x 500ms windows, amplitude threshold > 8<br>Run 2: 0-125Hz, 11 traces x 500ms windows, amplitude threshold > 8<br>Run 3: 0-3Hz, 31 traces x 1000ms windows, amplitude threshold > 1<br>Run 4: 0-5Hz, 11 traces x 1000ms windows, amplitude threshold > 4<br>Run 5: 6-12Hz, 11 traces x 500ms windows, amplitude threshold > 5 | |
| **Visual checking** | **Missing shot interpolation**<br>Sorting to Common shot gathers<br>1st small gap interpolation in the F-K domain using patches of 500 ms x 21 traces<br>2nd big gap interpolation in the F-K domain using patches of 300 ms x 125 traces | |
| (**Output difference, Stack**) | **Linear Radon Filtering**<br>Passing P range -0.35 to 0.7ms/m | |
| (**Output difference, frequency spectrum, Stack**) | **Deghosting in FK domain**<br>Zero padding to left and right of shot gathers<br>Extrapolation to left and right of shot gathers<br>Receiver-side deghost, patch 2000 ms x 21 traces, reflection coeff -0.95, Regularization 0.05<br>Shot-side deghost, patch 1400 ms x 240 traces, reflection coeff -0.95, Regularization 0.1 | **Stacking velocity analysis**<br>500m CMP interval for every line |
| **Visual checking, frequency spectrum** | **Signature Estimation**<br>Align water bottom<br>trim corrections base on max cross-correlation peak of traces<br>Stack | |
| **Visual checking, frequency spectrum** | **Designature operator estimation**<br>Design matching filter between water-bottom derived signature and a targeted band limited Zero-phase Ormsby wavelet 4-8-40-120 | |
| **Output difference, Stack** | **Designature Application** | |
| **Output difference, Stack** | **Low-Frequency attenuation**<br>Ormsby low-cut filter 3-7 Hz | |
| **Visual, Frequency spectrum** | **Inverse Q correction (phase)**<br>Q =120 below seabed, reference frequency = 50 Hz | |
| **Residual move-out, stack** | **Isotropic Pre-Stack Time Migration, then velocity refining every 250 CDP (~1.5km)**<br>Offset regularization | |
| **Visual** | **Radon Demultiple Model**<br>Parabolic radon transform from -200ms to 500ms (reference offset= 2500m) with an AGC wrap of 500ms<br>Left mute (Decreasing from 100 to 30 ms depending on iterations) | |
| **Output difference, Stack** | **Multiple substraction**<br>Data - Radon Model | |
| **Output difference, Stack** | **10 CDP (~50m), Automated residual velocity analysis**<br>Auto-picking +- 10% velocity, every 25 ms<br>Pick smoothing (51 CDPs) | → - OUT: Pre-migration final demultiple data<br>- OUT: Stack & Migration RMS velocity |
| **Visual, Frequency spectrum** | **Inverse Q correction (amplitude)**<br>Q =120 below seabed, reference frequency = 50 Hz | |
| **Stack** | **Stack**<br>Angle mute 48deg<br>Stack<br>Waterbottom mute | → - Processed Migration stacks |

**Figure 2 Time domain reprocessing sequence and parameters applied to the SALTFLU dataset**

Processing is performed using the REVEAL software from Shearwater Geoservices. After loading, raw data are checked for integrity using header values (Field File Identification Number (FFID), shot point number, channel number) and cross-checked with the observer logs to assess FFID-shot point integrity. Near-trace plots are visually analysed to identify any problems with the source array (e.g., misfires, missing shots) and receiver channels (e.g., noise bursts, electrical interference). The acquisition setup did not record information on the positioning of the streamer, preventing accurate location of the common mid-points (CMPs). Instead, we use the ship positioning GNSS data and project the streamer along the smoothed ship track, resulting in CMP locations that roughly honour the ship track and likely feathering of the streamer. The CMP binning is done using 6.25 m bins along the profile, giving a nominal fold of 60. After merging the navigation geometry into the trace headers, we subtract the recording time shift (40 ms), calculated as the average zero-offset time delay of the direct arrival (linearly extrapolated from the recorded data with a water velocity of 1511 m/s).

### 3.1. Noise attenuation

This stage aims to attenuate noise that could affect the deghosting and the estimation of the source signature. Firstly, data are resampled from 2 to 4 ms. A 3 Hz/30 dB-110 Hz/96 dB Butterworth bandpass filter is applied in the frequency domain (Figure 3b). The low- and high-cut values are chosen after analysis of the octave panels to ensure no elimination of signal (Figure 3a). The 3 Hz low-cut attenuates the very low frequency (1-3 Hz) part of the swell noise caused by pressure variations along the streamer (Bedenbender et al., 1970; Dondurur, 2018). The 110 Hz high-cut filter is an anti-aliasing filter for later re-sampling to 4 ms.

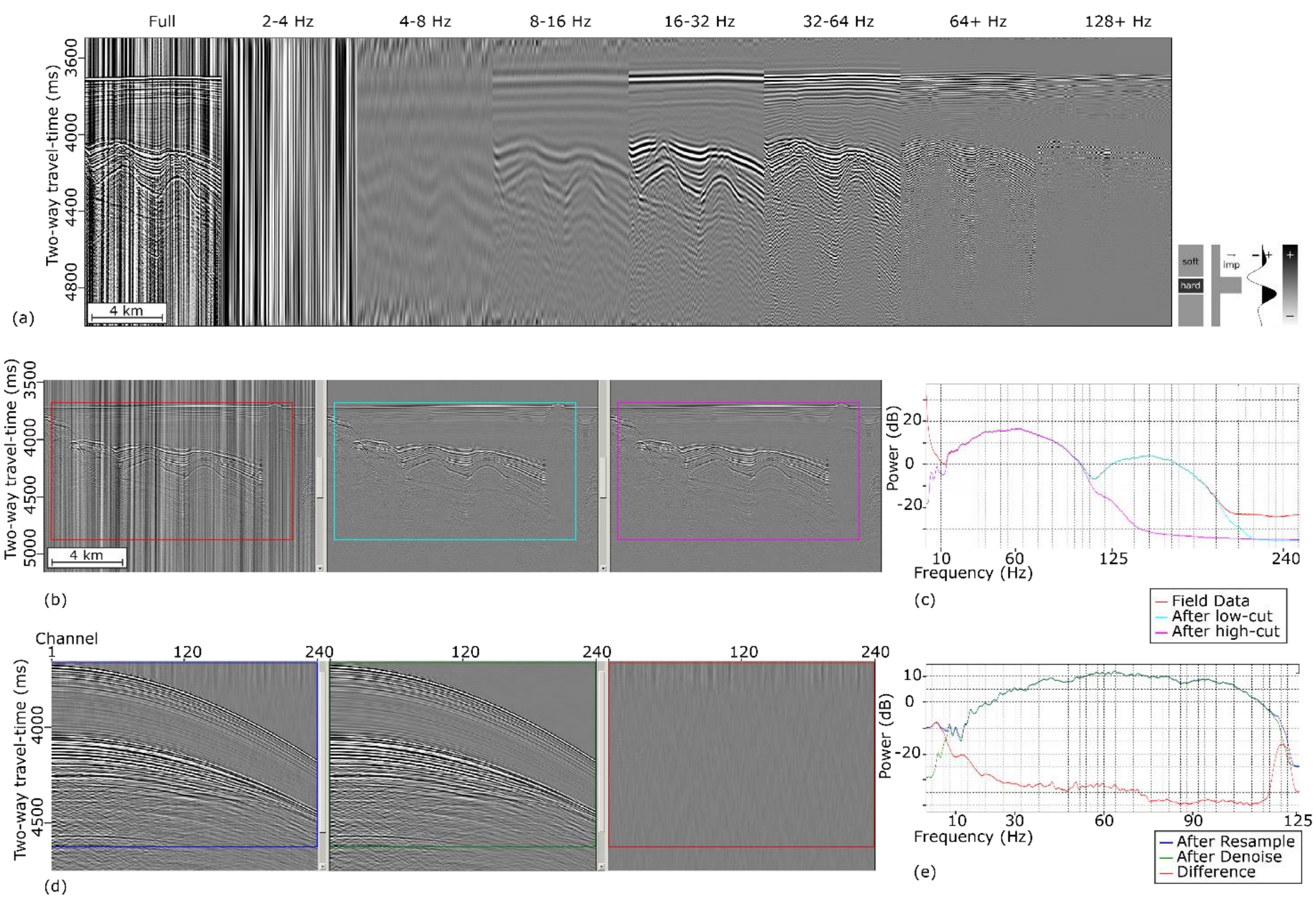

**Figure 3 Seismic sections of part of the SF05 line illustrating the results at different steps of the noise attenuation stage (a) Frequency panels at different octaves of raw field data; (b) Input near-trace-plot (left), outputs after low-cut filter (middle) and high-cut filter (right); (c) Frequency spectra corresponding to (b); (d) Input shot gather after resampling to 4 ms (left) versus after linear low-frequency noise removal steps (middle), and difference panel between the two (right); (e) Frequency spectra corresponding to (d).**

The remaining low frequency part of the 'swell' noise is addressed by several iterations of frequency-space (F-X) domain prediction filtering (Hornbostel, 1991; Schonewille et al., 2008). We perform up to 5 iterations on common-receiver gathers, followed by 5 iterations on common-shot gathers. The first two iterations target broad noise bursts at all frequencies, while later iterations target local noise bursts with frequencies between 0-3 Hz, 0-5 Hz, and 6-12 Hz respectively. The total number of iterations depends on the amplitude of the 'swell' noise for each individual profile. The F-X prediction filtering has the secondary effect of attenuating amplitude spikes present in the dataset (e.g., from electrical interference).

After this initial filtering and resampling, missing shots and traces removed during shot and channel edits (Section 2) are interpolated on common-receiver gathers. This is performed by frequency-wavenumber (F-K) domain interpolation in two steps: 21 traces x 500 ms patches to target small gaps, then 125 traces x 300 ms patches to target larger gaps. Finally, forward and inverse linear Radon (τ-p) transforms are performed on shot gathers to further reduce the remaining 'swell' and random noise (Figure 2).

### 3.2. Deghosting and source designature

Deghosting is performed on shot gathers in the F-K domain to attenuate the source- and receiver-side 'ghost' effect resulting from sea surface reflections. Shot gathers are transformed into the F-K domain, where the notch location varies as a function of frequency and horizontal wavenumber, illustrating that the ghost time-delay is dependent on the emergent angle (Amundsen, 1993; Day et al., 2013). The optimal deghosting operators for each frequency and wavenumber are estimated through a least-squares inversion scheme, minimizing the error between the model and the recorded data, parameterized based on the acquisition and medium parameters (i.e. receiver depth, water velocity, sea surface roughness). An inverse two-dimensional (2D) Fourier transform is then performed to return the deghosted data in the time-space domain. The deghosting is window based to account for the non-stationary nature of the ghost in time and space. Prior to deghosting, shot gathers are padded by inserting 30 near-offset traces (362.5 m) and 10 far-offset traces (112.5 m) to prevent amplitude leakage from the forward and inverse transforms (Yilmaz, 2001). The receiver-side ghost is estimated in local patches (500 ms x 11 channels), and the source-side ghost is estimated in broader time patches (1000 ms x 240 channels). After deghosting, a further F-K filter is applied to remove artefacts that may be introduced during the deghosting, particularly at the edge of the gathers. A regularization parameter is applied (0.01 for the receiver side, 0.001 for the shot side) to prevent the amplification of low frequency noise (Wang et al., 2017; Denisov et al., 2018).

After deghosting, we apply a deterministic source designature to collapse the bubble pulse, sharpen the wavelet and zero-phase the data (Figure 4a). We follow the workflow of Sargent et al. (2011), where the far-field source signature is derived from the flattened and stacked water-bottom reflection (Figure 4d). A matching filter is then designed to convert the estimated far-field source signature to a zero-phase band-limited Ormsby wavelet (4-8-70-120 Hz). Using a single deterministic operator yields better sub-seabed imaging and low frequency recovery superior to probabilistic methods (e.g., surface consistent, spiking and predictive deconvolution) and modelled source methods (Sargent et al., 2011; Davison and Poole, 2015; Maunde et al., 2017). Converting the seismic data to zero-phase tends to improve temporal resolution and lowers interference between closely spaced reflectors, allowing for better delineation of the stratigraphy, and is a necessary pre-condition for Kirchhoff-type migrations (Sheriff and Geldart, 1995).

Finally, an inverse-Q filter (phase only) is applied immediately before migration to compensate for dispersion effects caused by seismic attenuation (Sams et al., 1997). Dispersion introduces a phase-shift with increasing propagation depth. Compensating for this is necessary to ensure that the wavelet remains zero-phase with increasing

propagation depth depth (Wang, 2006). A two-layer Q model is used, with $Q = 1000$ in the water layer (essentially non-attenuating). We choose a value of $Q = 70$ in the subsurface, based on a constant single value that best flattens the frequency spectrum and balances amplitudes with depth across all the profiles in the survey.

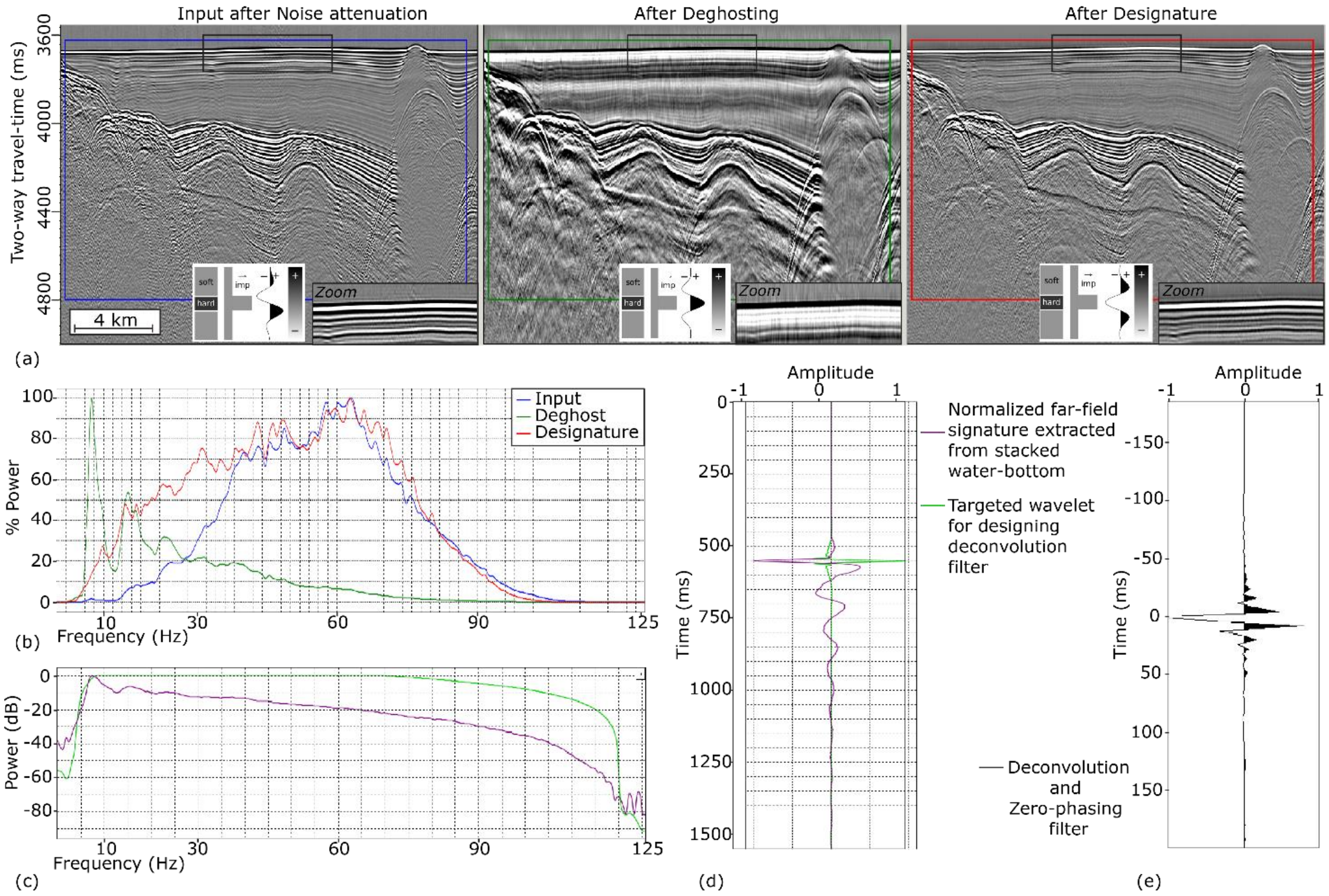

**Figure 4 Results from the bandwidth enhancement stage. (a) Near-trace gather before deghosting (left), after deghosting (middle) and after source designature (right); (b) Frequency spectra corresponding to (a); (c) Far-field source signatures estimated from the**

### 3.3. Multiple attenuation

The water depth exceeds 1500 m in the survey area and the free surface 'water bottom' multiples generally arrive later than most of the reflected primary signal at the target depths. High amplitude short period internal multiples are, however, observed throughout the survey, generally linked to the strong impedance contrasts associated with the salt (Fig. 5). We therefore attempt to attenuate both long and short period multiples using moveout discrimination techniques.

Before migration, we perform multiple attenuation in Radon domain (Figure 5). Normal moveout (NMO) correction is performed using the RMS velocity model (Section 2.4.1) to flatten primary energy, and the CMP gathers are transformed into the Radon domain using a least squares parabolic Radon transform (Sacchi and Porsani, 1999). The Radon domain multiple model is defined by a time-varying inner moveout mute (Figure 2). A top mute is applied at 200 ms below the water bottom to avoid attenuating primary energy in the shallow Plio-Quaternary (largely free of internal multiples). We then perform the inverse Radon transform on the multiple model and subtract from the original data.

We perform a second pass of Radon multiple attenuation post-migration, with a similar workflow to the pre-migration demultiple. For the depth domain processing (Section 2.4.2), the depth migrated CDP gathers are first

245 converted to time domain using the time migration velocities, the Radon demultiple is applied, and the CDP gathers are converted back to depth domain using the same velocity field.

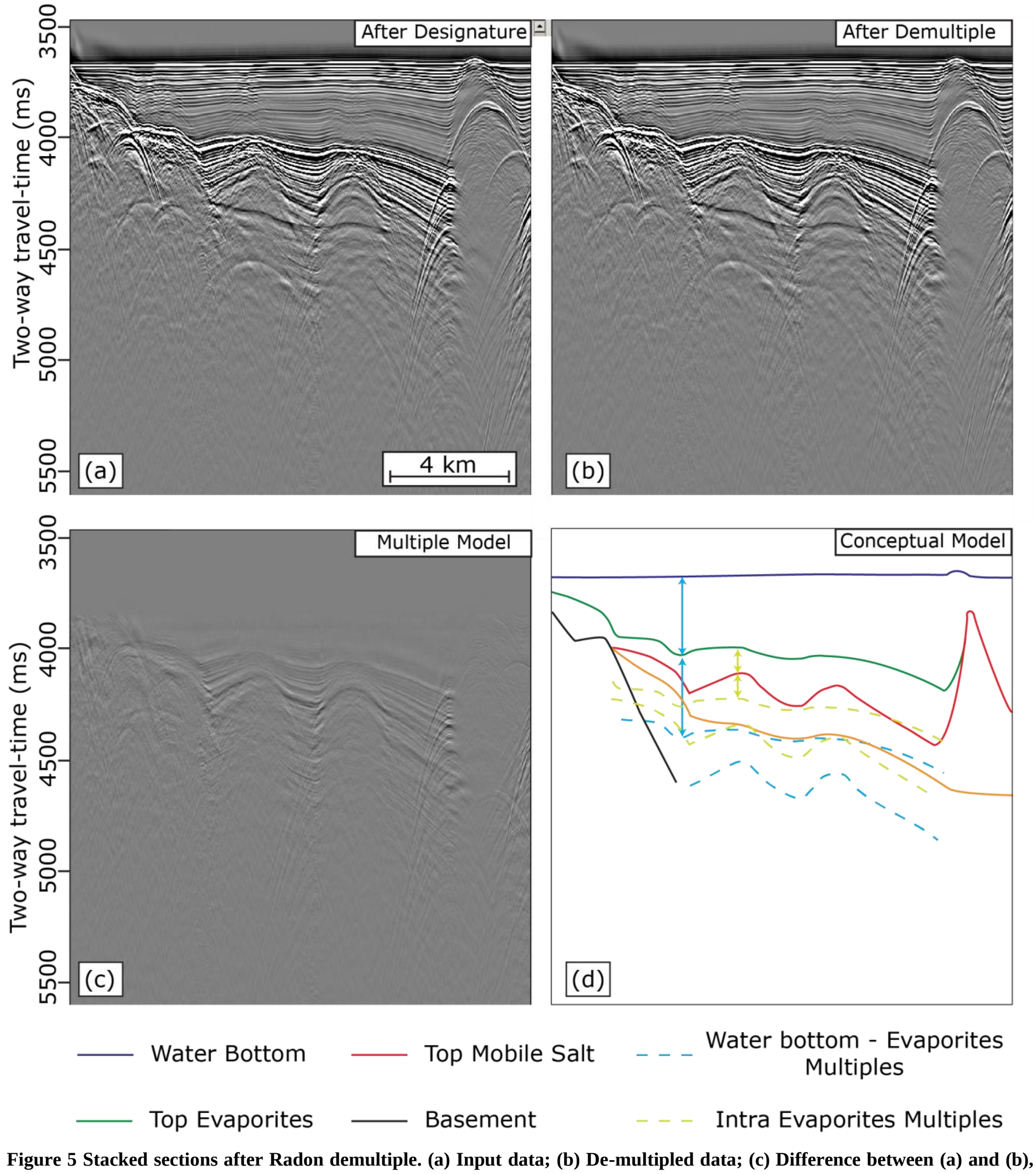

250 **Figure 5 Stacked sections after Radon demultiple. (a) Input data; (b) De-multipled data; (c) Difference between (a) and (b), i.e., the modelled multiples; (d) Sketch cartoon of the major multiple generating horizons and corresponding multiples. Due to the water depth water-bottom (blue) related multiples are not superimposed on the primary reflections in the target geology and are not an issue for the deep-water Algerian basin. Most multiples are observed below the Top Evaporites (green).**

### 3.4. Iterative migration and velocity model building

#### 3.4.1. Time domain

Our approach to velocity model building in the time domain involves picking RMS velocities on semblance panels on migrated CMP gathers. We perform several iterations of migration velocity analysis, gradually refining the velocity field.

A sparse initial velocity model is picked from semblance analysis of unmigrated CMP gathers every 500 CDP (~3km). This initial velocity model is primarily used as a stacking velocity to quality control intermediate processing stages, and for the initial pre-stack time migration.

We use a Kirchhoff pre-stack time migration to image the data. This choice is dictated by a limited computing resources and the flexibility of this algorithm to velocity errors. Before migration, the data are regularised into evenly spaced offset bins of 50 m using a partial NMO correction. The migration aperture is parameterised by analysing the migration impulse response (Appendix A). Aperture is broadened from 1500m at 1500ms to 3500 m at 5000ms. After the migration, we perform Radon domain multiple attenuation (Section 2.3) with broad parameters (minimum moveout -100 ms, maximum moveout +300 ms) to avoid attenuating primary reflections for the following velocity analysis.

We perform migration velocity analysis on the migrated CMP gathers every 250 CMP (~1.5 km). The migration followed by velocity analysis is repeated three times until the primary reflections are flattened on the migrated gathers.

#### 3.4.2. Depth domain

Depth domain processing aims to estimate an interval velocity model and produce depth migrated seismic images. Similar to the time domain model building, we use an iterative approach by repeatedly performing velocity analysis on migrated gathers, gradually refining the velocity field. Our criteria for a successful velocity model are that the primary reflections are flat after depth migration.

We use a Kirchhoff pre-stack depth migration to migrate the data in depth domain. The input data to the migration are the regularised CMP gathers (after Radon demultiple and phase-only inverse-Q filtering) as were used in the pre-stack time migration (Section 2.4.1). The parameters used for travel-time computation are detailed in Appendix B.

The initial velocity model is built using a water velocity (which we assume to be constant) and sediment velocities derived from the time domain velocity analysis (Section 2.4.1). The water velocity is estimated by performing a range of constant-velocity depth migrations (1450-1550 m/s). We choose a constant water velocity of 1525 m/s for the whole SALTFLU survey, which best flattens the water bottom reflection across the different profiles and at different water depths. To derive the interval sediment velocities, we first flatten the time domain velocities along the water bottom. The RMS velocity models in time domain are then converted to interval velocities in depth domain using a Dix conversion (Dix, 1955). The resulting interval velocities are Gaussian smoothed (100 m x 500 CDP) before shifting back to the original water bottom depth. An initial depth migration is performed using this initial velocity model and the input gathers to the final pre-stack time migration.

Following this initial pre-stack depth migration, we obtain velocity updates using a global tomography procedure comprising automated offset-dependent residual moveout picking, ray tracing and travel-time tomography. The velocity update in each iteration is limited to -10% and +20% of the input velocity, respectively. The automated picks are quality controlled by visually checking that the picks correlate with major reflectors, eliminating picks with

low trace-to-trace correlation and eliminating picks with anomalously high moveout correction. Travel-time tomography is then used to compute model updates from these edited picks. The parameters used for tomographic updates for each velocity model building iteration are detailed in Appendix C. We consider that the velocity model is good enough when most reflectors appear flat on migrated gathers. The first two iterations are performed only for the post-salt sediments, updating only between the interpreted water bottom and the top salt horizons. Initially, the top salt horizon is interpreted from the time domain data and converted to depth using the initial velocity model. The picking of the top salt horizon is refined for each velocity updates to fit the new depth image and limit the tomographic iteration to the post-salt.

The sediment model is then flooded with salt velocity from the top salt horizon to the base of the profiles. The velocity used for the flooding is chosen by migrating profiles with the salt flood models with a range of salt velocities and observing the flatness of the base salt reflector. We find that a salt velocity of 4300 m/s best flattens this reflector for survey profiles. The transition from the sediment velocity to the salt velocity is Gaussian smoothed (50 x 50 m) to enable projected rays to pass through the sharp velocity contrast during travel-time computation for the migration and tomography. The base salt horizon is interpreted on the depth migrated stacks after salt flooding, and a new velocity model is built with a salt velocity flooding that stops at this newly picked base salt. The pre-salt is also flooded with a velocity gradient starting from the base salt horizon. We test the velocity gradient with a range of starting velocities and velocity gradients and assess the overall flatness of reflectors in the pre-salt. After testing, the best results were obtained with a gradient of 2700 m/s + 1.2 (m/s)/m. Finally, two more iterations of tomographic velocity updates are performed following the pre-salt flooding. After the final depth migration, Radon domain demultiple is performed as previously described for the time domain processing (Section 2.3).

### 3.5. Post-migration processing

After the final migration, residual moveouts are automatically picked every 4 CDP (~20m) using semblance velocity spectra. The residual moveout analysis is limited to 5% change, and the resulting picks are smoothed spatially (25 ms x 200 CMP). The residual moveout field is applied to the migrated CMP gathers before stacking. Additionally, we apply an inverse-Q filter (amplitude only) to compensate for the frequency-dependent amplitude attenuation caused by seismic absorption. The Q model is the same as was previously for the phase-only inverse-Q filter (Section 2.2). We then apply an inner angle mute (at 1.2°) to target near-offset residual multiples and an outer angle mute (at 48°) to remove far-offset refractions and data affected by wavelet stretching. The CMP gathers are stacked, and a low-cut frequency filter is applied (Ormsby, 500ms, 4-12 Hz) in a short 400ms window below the seabed to target residual bubble energy (Figure 6).

## 4. Results

The initial bandpass filter eliminates the lowest frequency ‘swell’ noise and prepares the data for resampling from 2 ms to 4 ms by removing high frequency data (>125 Hz) that would become aliased (Figure 3b). The filter parameters are designed based on visually inspecting octave panels showing different frequency bands of the raw field data (Figure 3a). The 2-4 Hz octave panel is dominated by ‘swell’ noise and can be filtered out without losing significant primary signal. The 125-250 Hz octave panel shows that there is primary signal above 125 Hz. This implies that some high frequency signal is lost by resampling. The data are resampled both for computational efficiency and to ensure that any spatially aliased high frequency signal is removed before further processing stages, meaning that no

trace interpolation is needed for 2-D transforms in channel domain. This results in a considerable decrease in computing time for the most computationally intensive processing steps (e.g., deghosting, Radon de-multiple and pre-stack migration), at the cost of removing some high frequency primary signal. We consider this to be an acceptable trade-off for this study because the main objective is to better image the base salt, primarily by recovering the low frequency part of the data.

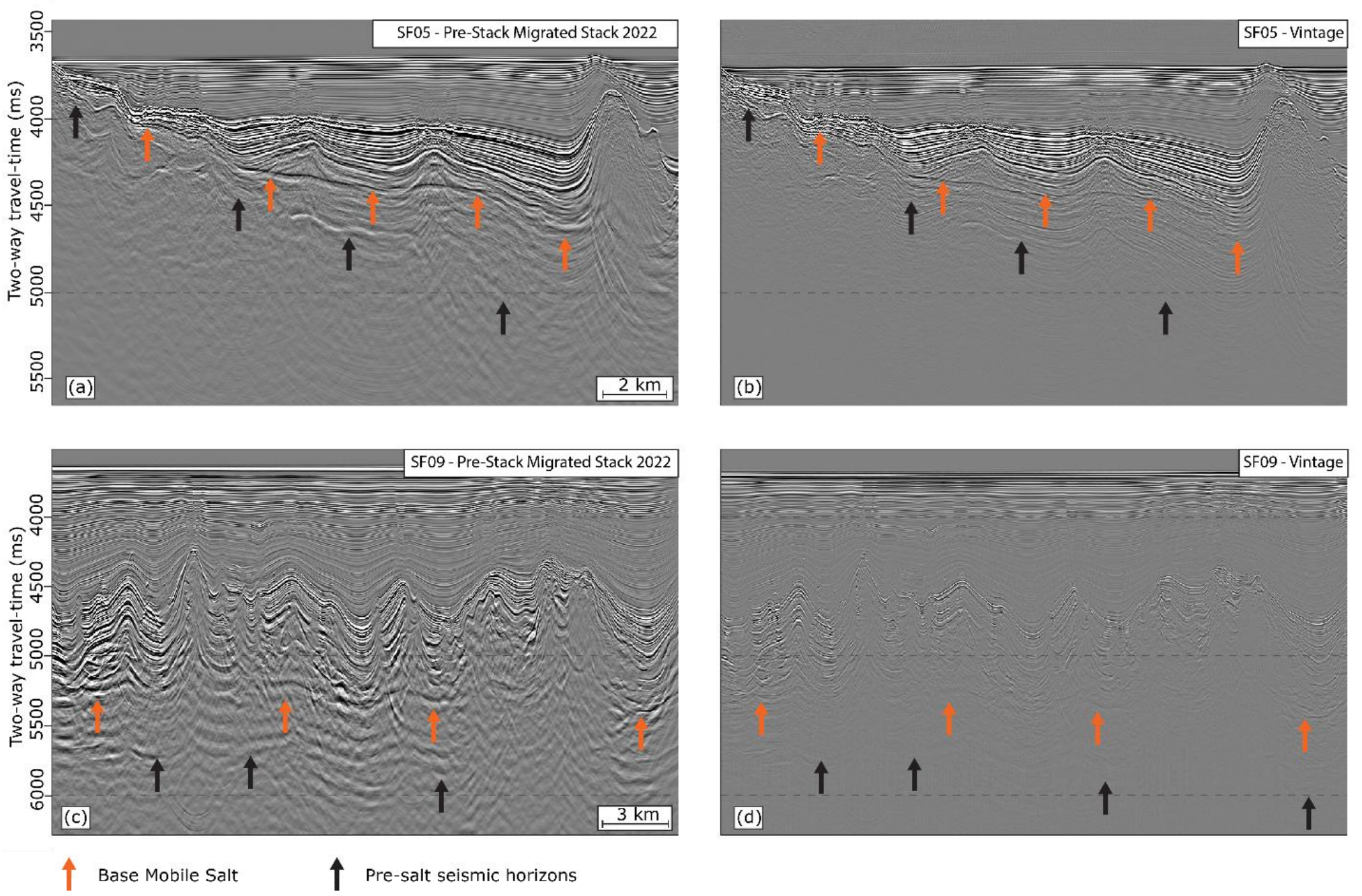

**Figure 6 Comparison between the newly re-processed sections (left; a and c) and the vintage sections (right; b and d) in time domain, for sections of lines SF05 (a and b) and SF09 (c and d). Bandwidth enhancement and Radon demultiple allow better imaging of the base salt horizon (orange arrows) and pre-salt horizons (black arrows).**

Following F-X prediction filtering and linear Radon filtering, most of the low frequency noise is removed, without attenuating primary signal or introducing artefacts (Figure 3d and 3e). Some of the noise remains between 8 and 16 Hz, but it could not be eliminated at this stage without filtering out primary signal and is cancelled out after stacking.

The deghosting sharpens the seabed and enhances the lower frequencies, highlighting the base salt, but also boosting the residual ‘swell’ noise and the low frequency bubble pulse (Figure 4a), with peaks at the harmonics of the bubble pulse frequency at 7 Hz, 15 Hz and 23 Hz (Figure 4b). The bubble oscillation is contained within the far-field source signature estimated from the stacked seabed reflection (Figure 4d). The matching filter between the far-field source signature and the targeted band-limited and zero-phase Ormsby wavelet (Figure 4c and 4d) successfully sharpens the source wavelet and eliminates the bubble oscillations (Figure 4a) thereby improving the balance of the frequency spectrum (Figure 4b).

Radon demultiple attenuates the internal multiples observed below the salt but does not completely eliminate them (Figure 5). The multiple model also contains a weak component of the primary signal, but this ‘primary leakage’ is considered acceptable as it does not significantly alter the amplitude of the reflectors after stacking.

Based on RMS velocities and dominant frequencies ranging from 1511 m/s and 60 Hz at the seabed to 5500 m/s and 10 Hz at the pre-salt, the nominal maximum vertical resolution of the new results varies from approximately

6.3 m to 137 m. Based on the same velocities, the legacy time migrated stacks display a slightly higher dominant frequency of 65 Hz, hence they have very slightly higher vertical resolution (5.8 m) at the water bottom. The legacy processing, however, did not include deghosting, a robust source designature or zero-phasing, so the effective vertical resolution of the legacy data is in fact much lower than this nominal resolution. The amplitude content of the new pre-stack time migrated stacked sections appears better balanced than the vintage dataset, particularly for the low frequencies (Figure 6a and 6b). The wider bandwidth of the reprocessed data better highlights the impedance contrasts and improves the interpretability of the data, particularly for the base salt and the pre-salt reflectors. The reprocessed new dataset displays lateral amplitude variations within the Plio-Quaternary and the Upper Unit that were not clearly visible before (see Section 4.3.2).

The pre-stack depth migration corrects for geometrical distortions linked to the strong lateral velocity variation between the evaporites and the encasing sediments (Figure 7). The ‘pull-ups’ of the pre-salt beneath the salt structures seen in the time migrated section are flattened after depth migration with an average velocity of 4300 m/s. Tomographic updates performed after salt flooding resolve velocity variation within the salt. The pre-salt reflectors are not always clearly visible on the final sections, thereby limiting the reliability of the velocity model beneath the salt.

## 5. Discussion

### 5.1. Limits of the pre-processing strategy

The chosen pre-processing strategy is determined by the computing costs and our ability to separate noise from primary signal. The time resampling from 2 to 4 ms eliminates the frequencies above 125 Hz and attenuates the frequencies from 90 Hz. This is due to the smooth roll-off of the applied filter, that is necessary to avoid ringing. By doing so, the highest resolution part of the data is lost but this allows a significant reduction of computing times. Resampling divides by two the size of the dataset and lowering the Nyquist frequency to 125 Hz filters out frequencies that would otherwise become aliased after resampling. That way, spatial transforms (e.g. F-K, Tau-P and Radon filters) can be performed without further trace interpolation to unwrap the aliased energy. This avoids multiplying the size of the dataset by at least a factor of 2, therefore the resampling reduces data size by 4, improving the computing times. The aim of this reprocessing is to improve the image of the evaporites and the salt generally lying below a thick Plio-Quaternary cover, therefore we consider this resolution loss an acceptable trade-off. A future study focusing on the shallower sub-surface could benefit from the preservation of these higher frequencies by processing the dataset at 2 ms sample rate.

During the acquisition of SALTFLU, the source array and streamer were towed at approximately 3 and 4 m below the sea surface, respectively. Deep streamer towing increases the operational weather window, reduces noise, and increases signal penetration (Carlson et al., 2007). However, a time-delayed reflection from the sea surface (a so-called ‘ghost’) can interfere destructively with the recorded signal and limit the resolution of the data (Schneider et al., 1964). These ghost reflections introduce notches in the spectrum of the recorded wavefield which particularly attenuate the low frequency component (Yilmaz and Baysal, 2015). For a water velocity of 1511 m/s at the surface (based on previous water velocity measurements and direct arrival slopes), the first frequency notch should appear around ~252 Hz for the source-side ‘ghost’, ~189 Hz for the receiver-side ‘ghost’, and ~108 Hz for the combined source and receiver ‘ghost’.

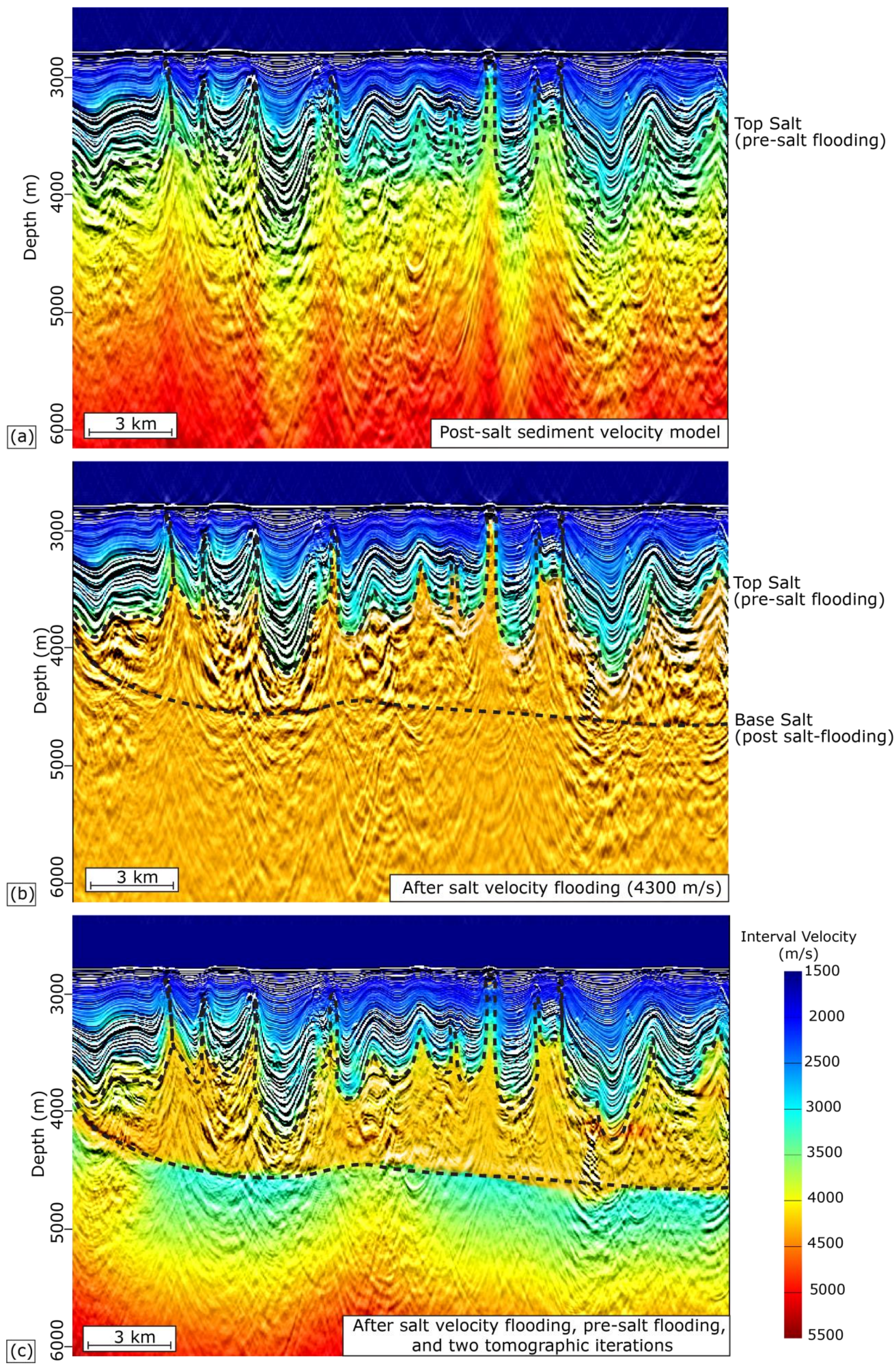

**Figure 7 Pre-stack depth migration sections and velocity models after 3 iteration of tomographic velocity updates in the post-salt (before salt velocity flooding) (a); after salt velocity flooding until the bottom of the sections in order to pick the base salt horizon (b); and after salt velocity flooding between top and base salt horizons and two iterations of tomographic velocity updates post flooding (c).**

In the data, the 'ghost notch' varies between approximately 118-150 Hz on near-trace gathers (offset 100m) depending on the lines, and it gradually shifts towards higher frequencies along channels until it disappears above filtered frequencies after the 10th receiver (offset 325 m). Such values broadly correspond to the combined source and receiver 'ghost' and might reflect a varying depth of the towed source or poor depth control along the streamer during the acquisition. With a Nyquist frequency of 125 Hz after resampling, the source- and receiver-side 'ghost' notches are mostly outside the data bandwidth (Figure 3c), but deghosting still helps to extend the lower part of the bandwidth and removes the interference of the 'ghosts' with the primary wavelet in time domain (Raj et al., 2016). A deeper towing depth would have been desirable to improve the recording of the lower frequencies and to attenuate the 'swell' noise, which is particularly strong and dominates the low frequency part of the data. It could not be fully eliminated pre-stack without attenuating some primary signal. The best results were obtained with iterative F-X prediction filtering, which better preserved the low frequency primary signal compared to the low-cut filter applied to the vintage dataset, but it required many iterations with different sliding windows on both common-shot gathers and common-receiver gathers. Other approaches tested include F-K filtering and time-frequency filtering, but these were either unsuccessful at separating the noise from the primary signal, or too computationally expensive. Tau-P domain filtering is also an efficient way to separate noise from primary signal (Basak et al., 2012). Here, we apply a shot domain forward and inverse Tau-P transform, with a limited slowness range, without further muting in the Tau-P domain. The 'swell' noise is incoherent enough to not sum constructively during the transform and is therefore well attenuated. The remaining noise embedded in the data is low amplitude enough to ensure a stable deghosting operator estimation, but it is boosted after the deghosting stage and an additional denoising step is required after bandwidth enhancement (Figure 2).

### 5.2. Limits of the multiple elimination and accuracy of the velocity models

As the water depth exceeds 1500 m in the survey area, long period surface-related multiples are not superimposed on the target primary signal. Therefore, surface-related multiple elimination (SRME; Verschuur, 2013) is not applied, and all multiples (long and short period) are attenuated solely by move-out based methods. However, short period internal multiples are present and not fully eliminated (Figure 5). They are generated by strong acoustic impedance contrasts either within the Upper Evaporites, or between the water bottom and the Top Evaporites. They prevent the resolution of velocity variations below and within the evaporites and can be misinterpreted as primary signal. During the processing they are only partially attenuated by move-out discrimination on CMP gathers, likely because they are generated by a high velocity layer (the upper Messinian unit above the salt). With a maximum offset of 3100 m, they only yield a small move-out difference compared to the primary reflections. Other approaches for attenuating multiples include boundary-related internal multiple method (Verschuur and Berkhout, 1996), data-driven internal multiple removal method (Jakubowicz, 1998) and inverse scattering internal multiple removal (e.g. Araújo et al., 1994). The second method has been tested but it was not applied in the final flow due to significantly increased computational times (16 hours for 5000 shots, while the survey consists of more than 170 000 shots).

The presence of remnant multiple can lower the accuracy of the time and depth domain velocity analyses. The time domain velocity model is built through several iterations of semblance-based migration velocity analysis (with a pre-stack Kirchhoff time migration, and Radon multiple attenuation). The remaining multiples and the out-of-plane reflections can generate high semblance values that can be wrongly picked. Consequently, automatic picks had to be

carefully quality controlled during the residual move-out analysis stage after both the time and depth migrations. Internal multiples generated within the Upper Evaporites can also be wrongly picked as the top salt horizon used for the velocity flooding. Two reflectors R1 and R2 could be interpreted as the top of the salt (Figure 8).

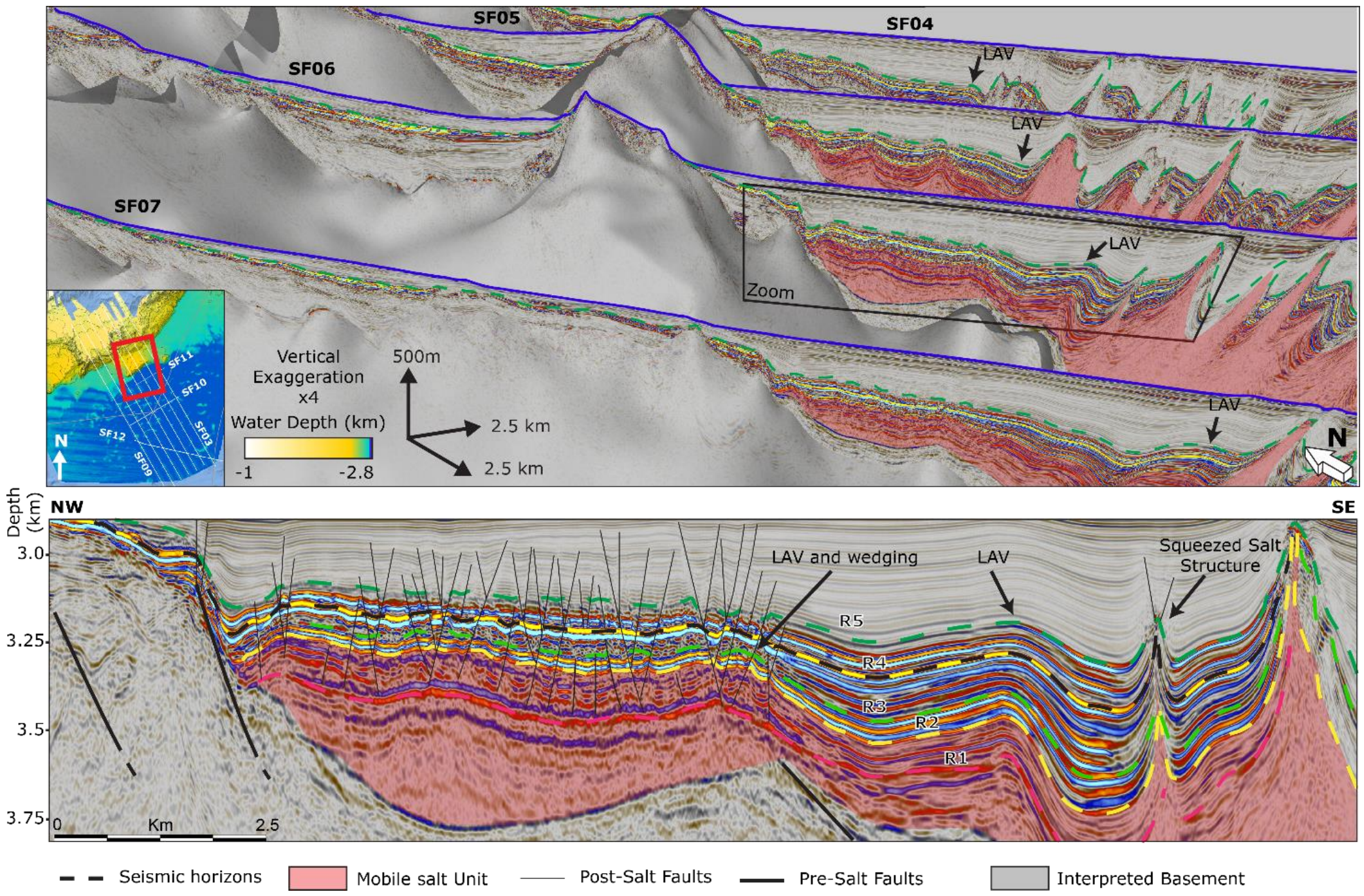

**Figure 8 3D view of the SALTFLU seismic profiles along the Balearic margin of the Algerian basin, with a zoomed section along line SF06 showing the post-salt Messinian seismic facies. R1, R2, R3 and R4 represents the four seismic horizons observed regionally in the Algerian basin within the evaporites. Vertical exaggeration x5. Positioning map from Figure 1.**

The top of the salt is difficult to follow laterally because of the complex salt structures and the presence of several reflectors within the salt that could represent residual internal multiples, out-of-plane reflections or the presence of internal clastic beds within the salt. The transition from the Upper Evaporites seismic facies to the 'transparent' salt seismic facies is gradual. The impedance contrast between the base of the Upper Evaporites and the salt is not clearly marked. Based on the strength of the amplitude and the lateral continuity of the reflectors, R2 seems to be the most likely horizon marking the top of the ductile salt. However, despite the likely presence of internal multiples below R2, some reflectors are believed to represent geology. We interpret them as intra-salt reflectors generated by the presence of clastic beds with the mobile salt. Intra-salt reflectors have already been observed in the eastern Mediterranean, where they lower considerably the average salt velocity to ~4260 m/s (Feng and Reshef, 2016). On CMP gathers, these reflectors align at a velocity of ~3250 m/s and would be over migrated if salt flooding was performed from R2 (Figure 9). To preserve them from the post-migration Radon demultiple, the salt flooding is performed starting from reflector R1. This choice has a strong control on the velocity used for the flooding to flatten the base salt (a lower top salt implies a higher flooding velocity to flatten the base salt), the resulting geometry of the salt structures on the stacked sections, and the depth of the base salt and pre-salt units. A better identification of the top salt could be guided by drilling through the UU until the top of the salt in the Algerian basin.

The final interval velocity models used for Kirchhoff depth migration are sampled every 12.5 meters laterally and 5 meters vertically. Tomographic updates were considerably smoothed and regularized to avoid velocity 'bulleyes'.

In the absence of nearby wells, the only quality check on the interval velocity model is whether the reflectors in common reflection point gathers are flat or not after migration. However, considering the limited offset of 3.1 km compared with the depth of the target and the strong velocity variations, the gathers get flattened under a substantial range of velocities. This implies that there are large uncertainties in the velocities, particularly for the evaporites and the pre-salt, where the residual internal multiples and the lack of strong pre-salt reflections make it difficult to constrain the velocity model.

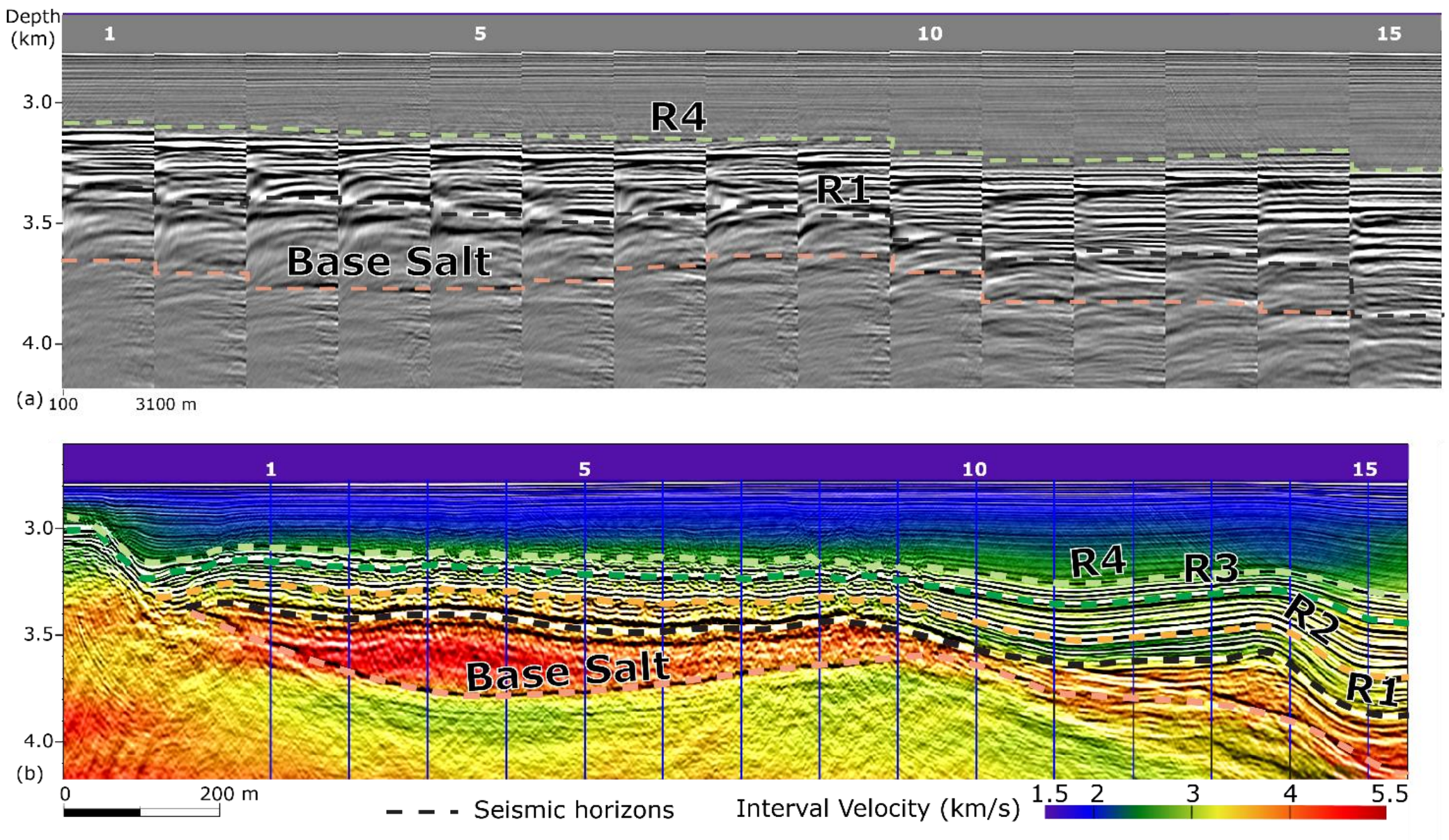

**Figure 9 A section of line SF06 after pre-stack depth migration: (a) common reflection-point gathers with interval velocity overlay; (b) a single migrated offset plane (offset 1063 m) with interval velocity overlay (positions of gathers in (a) marked). The velocity of the upper evaporites increases downward from 2500 to 3500 m/s. The mobile salt unit was flooded with 4200 m/s. Two rounds of tomographic velocity updates were performed in the pre-salt sediments after salt velocity flooding.**

The ray-based Kirchhoff migration is not the best suited strategy for salt imaging du to (Leveille et al., 2011): (i) a limited ability to handle steep and large velocity variations which requires a smoothing velocity model and limit the quality of the output images and (ii), the apparition of migration “swing” artifacts in poorly illuminated zones under or near salt bodies due to limitations of the assumption of the ray-based scheme. More advanced migration techniques (e.g., anisotropic migration or reverse time migration) have not been tested. They are more costly and are accurate with sufficiently detailed velocity models, thus they require corresponding higher resolution velocity estimation techniques (e.g., waveform inversion or anisotropic model building; Jones, 2015). These techniques require dense and long-offset recording of the reflected wave field, a good knowledge of the geology and petrophysics distribution within the medium, with ideally nearby well control, and good computational power (Virieux and Operto, 2009). SALTFLU data is sparse (~6 km spacing between the lines), with no nearby wells, and the maximum offset recorded is ~3.1 km, while the local subsurface is known to include a highly deformed and deep (>4 km ) salt (Camerlenghi et al., 2018). For these reasons, the Kirchhoff migration is preferred (Yilmaz, 2001; Vardy and Henstock, 2010; Dondurur, 2018). In practice, it remains a flexible and robust approach where the velocity model is not well constrained and the objective is primarily the imaging of salt structures, not the subsalt imaging, but reverse time migration could potentially improve the imaging of the salt structures.

### 5.3. Area of interest within the central Algerian basin observed on the reprocessed data

The newly reprocessed data better image the pre-salt reflectors, allowing an improved understanding of the salt tectonic system from late Miocene to today in the central Algerian basin (Blondel et al., 2022), with a peak of contractional salt deformation at early to mid Plio-Quaternary. This peak could be linked to an important episode of shortening of the Tell-Atlas fold-and-thrust belt, along the Algerian Margin. A contemporaneous tectonic event has been also identified along the Balearic margin, with a regional unconformity, and the reactivation of pre-existing faults leading to the uplifting and the tilting of the Balearic slope (Blondel et al., 2022).

SALTFLU is one of the highest seismic resolution datasets available in the Algerian basin. It allows detailed study of the late Miocene seismic facies, the acoustic basement in the Formentera basin, and faulting within the Plio-Quaternary that could provide new insights on the geological evolution of the region. Combined with other neighbouring datasets, regional sections extending from the Mallorca shallow basins to the Algerian margin through the Formentera basin and the deep Algerian basin could be drawn.

#### 5.3.1. Mud volcanoes and fluid circulation

The presence of mud volcanoes in the Algerian basin has previously been suggested by Camerlenghi et al. (2009). In salt basins, shale diapirs are common in areas of shortening, which can pressurize and mobilize shale, as can vertical loading (Jackson and Hudec, 2017). Mud volcano systems have been recognized in the neighbouring Alboran sea, where they are sourced by pre-Messinian sediments (Aquitanian–Burdigalian in age; Sautkin et al., 2003; Blinova et al., 2011; Medialdea et al., 2012; Somoza et al., 2012). We observe several weakly reflective diapir contacts, with no underlying velocity pull-up but with a narrow dimming zone instead (Figure 10d). Migrating this structure with a high salt velocity result in over-migration artefacts, suggesting that either (i) they are not made of high velocity material, but could be made of low velocity mud; (ii) the structures are are out-of-plane with respect to the 2-D seismic profiles; or (iii) the salt is allochthonous and too thin to generate a pull-up. They could testify to an active mud volcano system, but the evidence is too poorly constrained to verify this hypothesis.

We observe several low amplitude anomalies with accompanying push-down and dimming along the Balearic margin (Figure 10a, 8b) and in the south-central Algerian foredeep (Figure 10c, 8d). In the Algerian foredeep we interpret these as gas chimneys, which supports the presence of an active fluid migration system. They are systematically located above a pierced UU, suggesting they need a migration path through the UU to escape vertically into the Plio-Quaternary. Somehow, they do not migrate further up dip along the Plio-Quaternary strata (Figure 10d), suggesting the potential presence of stratigraphic traps. It is not known if this gas is thermogenic or biogenic in origin. Previous studies in the Mediterranean sea have shown that cross-evaporite fluid flow through an hydro-fractured MU is likely, implying that the gas observed in the SALTFLU profiles could potentially be sourced from a pre-salt source rocks (Dale et al., 2021; Oppo et al., 2021). A thermal modelling study in the eastern Algerian basin from Arab et al. (2016) favours the pre-salt Oligo-Miocene sourcing, with the Messinian shale as a possible biogenic gas source. In the Algerian foredeep, seismic fluid indicators are only observed in the western part of the study area, which is characterized by relatively undeformed Plio-Quaternary deposits, and less deformed salt compared to the eastern part (Blondel et al., 2022). The absence of gas chimneys in the eastern part could be due to the absence of traps, and/or migration pathways, and/or the immaturity or absence of a source rock. If there were fluids migration but no traps, we would expect to see evidence of fluid escape features, such as pockmarks, at the seabed. Arab et al. (2016) conclude that discharge and accumulation of the pre-salt thermogenic fluids is limited to the Algerian margin or beyond the slope toe (60 km from the coastline), with some structural traps associated with the Quaternary north-verging thrust ramps.

Further eastward, the basin widens and the thrust ramps are located further away from the seismic data (Blondel et al., 2022).We speculate that the fluids may not be able to migrate far enough within the basin to be observed locally of the eastern lines. The western part of the Algerian basin also displays a higher heat flow than the eastern part (Poort et al., 2020), this spatial variation could have had an impact on the production of fluids, whether biogenic or thermogenic.

Along the Balearic margin, fluid indicators are observed locally when salt is absent (Figure 10a and 8b). The nature and the source of these fluids are also unknown. Previous studies link these seismic amplitude anomalies and the normal faulting on the Balearic margin to hydrofracturing induced by the fluid circulation (Urgeles et al., 2013; Wardell et al., 2014; Del Ben et al., 2018; Dale et al., 2021).

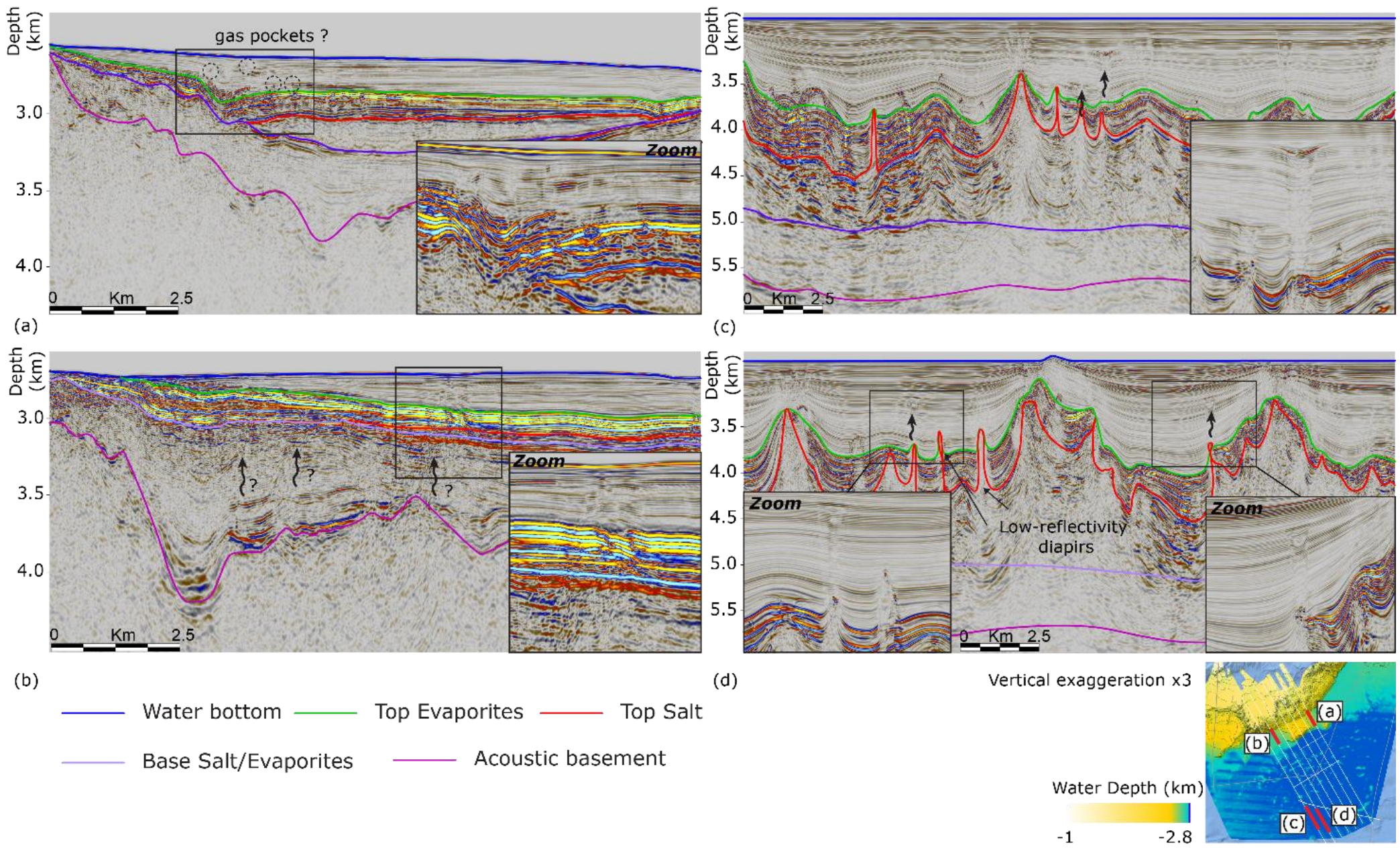

**Figure 10 Sections in depth of lines SF03 (a), SF08 (b and d), and SF09 showing amplitude anomalies and disturbed bedding that may indicate the presence of fluid migration. Arrows indicate areas of blanking and/or disturbed bedding, representing possible fluid migration pathways. Vertical axes indicate depth below the sea level. Vertical exaggeration x3. Positioning map from Figure 1.**

### 5.3.2. Seismic facies variations within the Messinian seismic units

Along the Balearic margin, the Upper Unit is interpreted as an evaporite-rich layer, with two subunits UU1 and UU2 separated by an intra-UU unconformity, and possibly, an intra-UU salt layer (Dal Cin et al., 2016; Camerlenghi et al., 2018). The reprocessed SALTFLU profiles show that the UU can be divided into more than two units (Figure 8). These units display a cyclicity in amplitude strength that could be linked to changes in lithology or depositional environment. The lowermost seismic unit of the UU, between reflectors R1 and R2, is characterised by relatively discontinuous medium amplitude reflectors embedded in a low amplitude seismic background 'matrix', with a velocity comparable with the overlying units (3000 +/- 250 m/s; Figure 8). This unit could represent a transition facies from the ductile salt (dominantly halite with poor clastic content) to more brittle Upper Evaporites, richer in clastic and gypsum evaporites. Many faults can be tracked up to the reflector R1 which attests to brittle behaviour (Figure 8). Whether it is part of the UU or the MU, this unit pinches out before the overlying units.

The presence of an intra-UU salt layer and of an intra-UU erosion surface is questioned. Reflector truncations are only visible on SF03 at the outlet of a canyon of the EBE, and are not visible elsewhere (Figure 8, Figure 11).

Velocity profiles do not show any high velocity layer in the UU (Figure 9). On the contrary, the average velocity of the UU is low: it increases downward, from ~2500 m/s at R4 to ~3250 m/s at R1. These values cannot be associated to high velocity evaporites such as gypsum and/or anhydrite (~5000-6000 m/s; Schreiber et al., 1973; Mavko et al., 2009) or halite. This could indicate a clastic-rich layer, with an increase in the proportion of low-velocity material, where the last unit of UU is predominantly made of clay, such as in the Unit 7 of the Levant Basin (Gvirtzman et al., 2017). The scientific wells that penetrate the topmost UU in the Algerian basin confirm the presence of a sequence of mud and marls, interbedded with evaporites (dolomites, gypsum and anhydrite) that are possibly rich in organic matter (ODP-975, DSDP-124 and DSDP-371; Ryan et al., 1973; Hsü et al., 1978; Comas et al., 1996).

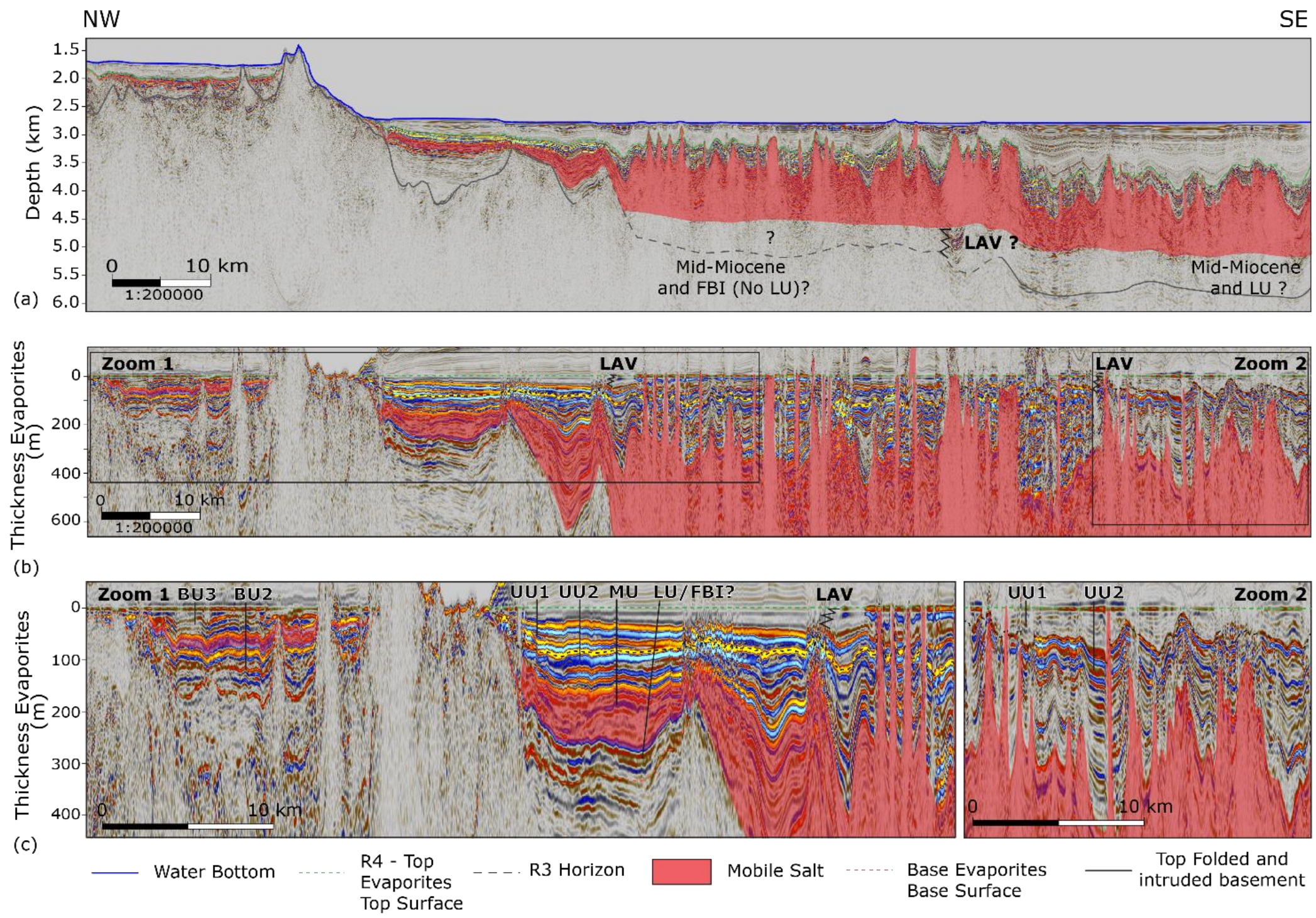

**Figure 11 Pre-stack depth migrated line SF09 before (a) and after flattening (b) along seismic horizon R5 (interpreted as the top of the Messinian evaporitic sequence). The two zoomed sections (c) along the flattened line illustrate the comparison between the Bedded Units of the Balearic promontory with the units of the Algerian basin, and the variation in seismic facies within the Messinian Upper Unit in the Algerian basin. Vertical exaggeration x8 for the original line, x20 for the flattened line.**

The Reflector R4, separating the UU1 from the UU2, is the strongest amplitude reflector observed within the whole survey (Figure 8 and 11). This regional strong impedance contrast could mark the part of the UU containing the largest proportion of evaporites and could record a climax for the UU stage.

The shallowest Upper Unit, between R4 and R3, exhibits lateral amplitude variation, both toward the Algerian and the Balearic margin, on all profiles (Figure 8 and 11). The reflectors of this unit lose their high amplitude expression toward the nearest Balearic margin and toward the deepest Algerian basin. This loss of amplitude could result from a variation in the acoustic impedance contrast, that could be related to a change in lithology or pore fluid content. This uppermost unit of the UU also thins down toward the margin. It nearly falls below the seismic resolution (Section 3) and is then expressed as a medium to strong amplitude reflection with negative polarity compared to the water bottom. It is not clear where it pinches out, but it appears to do so before the underlying unit, between R3 and R2.

This last underlying unit also displays a sudden lateral variation (Figure 8, Figure 11), pinching out towards the margin. The point where most reflectors onlap the R3 reflector could coincide with the point where the overlying unit falls below the seismic resolution. These sudden changes could indicate a different depositional environment, either due to local variation in accommodation space, sediment supply and/or salinity.

#### 5.3.3. Seismic expression of the Messinian Lower Unit in the central Algerian basin

The absence of a high amplitude LU below the salt, as it is expressed in the Provencal basin (Lofi et al., 2011), could indicate the absence of the LU in the Algerian basin; that the LU is much thinner than the vertical resolution of the data; or that there is an absence of strong impedance contrasts within this unit and within the pre-Messinian units. In the Levant basin, in the eastern Mediterranean Sea, the LU is also absent and the onset of the Messinian Salinity Crisis is marked by the Foraminifers Barren Interval (FBI), a 10s-of-meters thick, evaporite-free, shale unit that records the entire duration of the first stage of the crisis (Manzi et al., 2021).

It could be that the record of the onset of the MSC along the Balearic margin of the deep-water Algerian basin is similar to the record of the MSC in the Levant basin. Toward the southern Algerian margin, however, the expression of the pre-salt units changes laterally and displays a set of continuous low amplitude and low frequency reflectors that could record the LU (Figure 11). Such seismic facies have been associated with the LU in previous studies, where it would be made of 100-200 meters of plastic grey marls and gypsum based on nearby industry wells (Burollet et al., 1978; Medaouri et al., 2014). However, they seem difficult to differentiate from the Tortonian underlying units (Leprêtre, 2012). This could confirm a lack of impedance contrast between the two (nearby wells showed that the underlying Tortonian and Serravalian units was also made of gray marls, with intervals of pyroclastics sands or limestones; Medaouri et al., 2014) suggesting that in the Algerian basin, the LU, when present, is not as rich in evaporites that in the Provencal basin. In that case, the LU could be a lateral equivalent of the FBI, deposited preferentially in the deepest part of the Algerian basin. Alternatively, contemporaneously with the deposition of the LU in the Provencal basin, MU is already being deposited in the Algerian basin, and LU is a lateral equivalent of the lower MU (Roveri et al., 2019).

#### 5.3.4. Insights on the tectonic setting of the Balearic promontory

The new SALTFLU sections also provide improved imaging of the Formentera basin, which could allow better stratigraphic correlation with the Messinian units from the Mallorca depression and the deep-water Algerian basin and a better understanding of the tectonic history of the Balearic promontory (Figure 11 and Figure 12).

When aligning the seismic data to the top surface of the evaporites (Figure 11), it seems straightforward to correlate the BU3 with the UU above the salt. Following the interpretation of Raad et al, (2021), if the BU1/BU2 (contemporaneous in age) are equivalent to the Stage 1 primary Lower Gypsum, according to the correlation suggested by Roveri et al, (2019), the equivalent of the BU1/BU2 in the deep offshore is the thin FBI (likely below the seismic resolution). This could explain why a pre-salt evaporitic unit is not observed in the deep basin, while it is recorded by the BU1/BU2 in the intermediate depth basin.

Collapse structures are observed at the top of the BUs, in the deepest part of the Formentera basin (Figure 12). The top surface of the BU3 unit is generally conformable with the Plio-Quaternary in the Balearic promontory (Raad et al., 2021) and is erosive only on topographic highs, not in the depressions where salt is preferentially accumulated. This suggests that the depressions were never aerially exposed to erosion or dissolution. Instead, the collapse features could be due to the circulation of undersaturated fluids or diagenesis, similar to features observed in the Messinian

evaporites in the eastern Mediterranean (Bertoni and Cartwright, 2015) . Below the collapse structures, there seems to be a vertical dimming of the image that could be related to escaping fluids from the underlying pre-Messinian units or the basement itself attenuating the seismic amplitudes (Figure 12). Heat flow measurements on the Balearic promontory record anomalously low values that support the presence of groundwater fluid circulation (Poort et al., 2020).

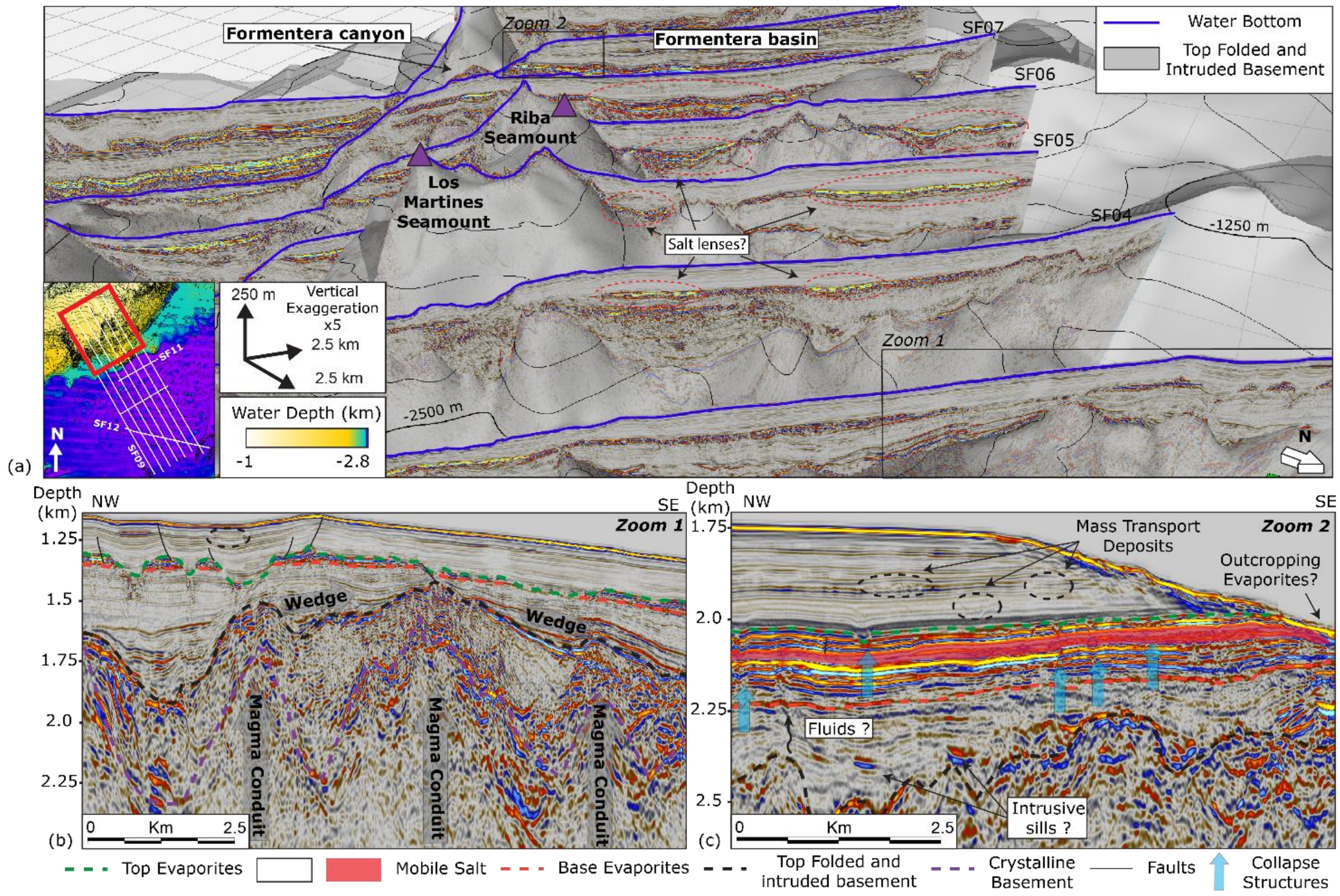

**Figure 12 3-D view of the SALTFLU seismic profiles along the Balearic promontory. Zoom 1 along line SF03 (zoom 1) shows the eroded and incised Messinian evaporites and the underlying volcanic basement, with onlapping wedges at both sides of the volcano. Zoom 2 along line SF08 shows the Messinian evaporites in the Formentera sub-basin, with several depressions at its top that suggest the presence of collapse structures, chaotic seismic facies in the Plio-Quaternary that suggests the presence of mass transport deposits, and disturbed signal below the Evaporites that could indicate fluid circulation. Vertical exaggeration x5**

In the reprocessed SALTFLU dataset, the pre-salt basement appears highly variable, containing several sharp and high-relief structures inferred to be of volcanic origin (Figure 12). These structures match the geometry and the seismic expression of igneous bodies in volcanic provinces, such as the Taranaki basin Infante-Paez and Marfurt, (2017). We interpret several magma conduits beneath the Miocene to the current sedimentary cover of the Formentera basin (Figure 12). They are overlain by a highly deformed unit, that is onlapped by pre-Messinian reflectors wedging towards the top of the magma conduits. These wedges indicate that the volcanoes were already present during the deposition of these pre-Messinian units. Extensive faults within the Plio-Quaternary cover suggest that the Formentera is still actively deformed (Figure 12).

## 6. Data availability

The SALTFLU dataset is available open access, for non-commercial use, on Zenodo (10.5281/zenodo.6908447).

Simon Blondel, & Jonathan Ford. (2022). Multi-channel seismic reflection profiles SALTFLU (Salt deformation and sub-salt fluid circulation in the Algero-Balearic abyssal plain) - Pre-Stack Kirchhoff Time & Depth Migration 2022 (Version 1) [Data set]. Zenodo. https://doi.org/10.5281/zenodo.6912214

## 7. Conclusions

We present the reprocessing of 2-D airgun seismic reflection data acquired offshore the Balearic Islands in the Algerian Basin in 2012. The goal is to improve the imaging of the Messinian evaporites and the overall basin architecture. The reprocessing strategy is designed to better image the pre-salt reflectors in an amplitude-preserving manner, attempting to overcome the challenges of imaging complex sub-surface geology using short-offset seismic data. This has been done using a 'broadband' processing strategy, multiple attenuation and an imaging approach that integrates geophysics and geological interpretation to iteratively build the velocity model. The resulting pre-stack migrated images, in depth and time, display improved reflector continuity and amplitude preservation, particularly for the pre-salt. The processing flow presented here could be applied to other short offset (<3 km) legacy airgun reflection seismic datasets in the Mediterranean Sea. The efficiency and efficacy of the workflow strongly depends on the original acquisition parameters: the limited offset of most vintage data (often <3 km) compared with the depth of the salt structures (the base salt lies in between 3.5 and 5.5 km) inhibits our ability to separate signal from noise and to accurately resolve the subsurface velocity distribution using moveout-based processing techniques. The reprocessed data reveal several fluid indicators, amplitude variations, salt structures and volcanic structures. These new results provide insights into the evolution of the under-explored Algerian basin and the Messinian Salinity Crisis. These outcomes highlight the value of reprocessing legacy academic seismic data, particularly when considering how challenging acquiring new seismic and borehole data has become in the western Mediterranean Sea.

## 8. Appendices

### Appendix A

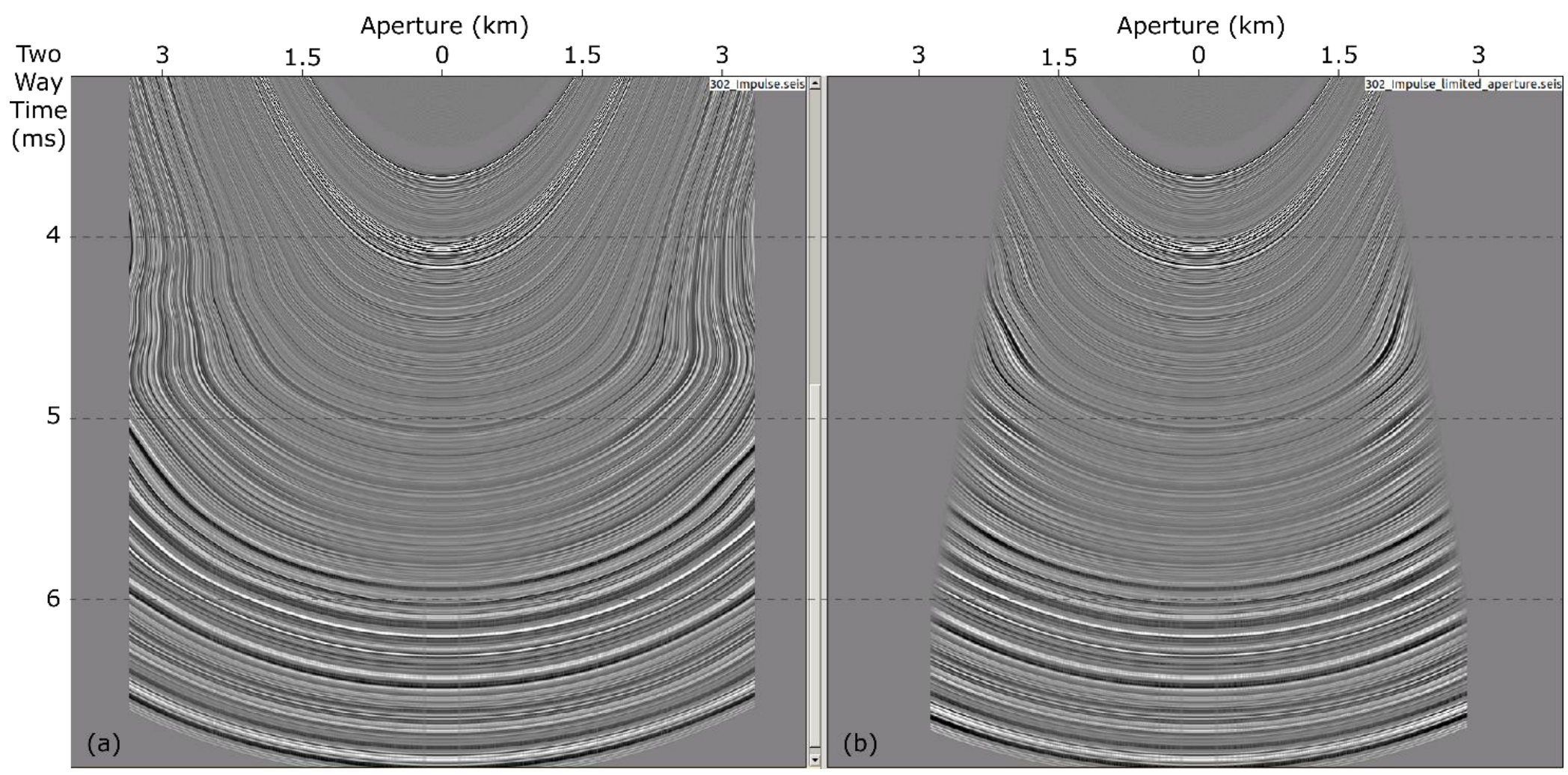

**Figure A1 Impulse response of the Kirchhoff Pre-Stack time migration in the deep Algerian basin, without (a) and with an angle aperture (b)**

**Appendix B**

660 **Table B1 List of the parameters used for the Kirchhoff pre-stack depth migration**

| PSDM Parameters | |
|---|---|
| **Ray Tracing** | |
| Gridding | 2 CMP x 5meters |
| Number of rat takeoff angles | 25 |
| Maximum takeoff angle | 60° |
| Stepper type | adaptive |
| Error tolerance | 0.0004 m |
| Mimimum step | 4 ms |
| Maximum step | 10 ms |
| Wavefront step | 10 ms |
| Coincident ray selection | minimum traveltime |
| Maximum ray separation | 100 m |
| Maximum ray normal divergence | 10° |
| Interpolate rays using | wavefront reconstruction |
| Maximum ray angle to vertical | 90 |
| Maximum travel time | 3500 ms |
| Maximum ray generation | 7 |
| Maximum shot time | 120 s |
| Gaps filling | Fast sweeping method for eikonal equations |
| **Kirchhoff prestack depth migration** | |
| Depth increment | 5 m |
| Aperture (Depth/Radius) | 1000-1500 /3250-3100/4500-4000 |
| Angle limit to aperture | 70° |
| Trace interpolation factor | 8 |
| CMP bin patch size | 3 |
| Antialisaing filter increment | 2.5 Hz |
| Antialisaing filter roll-off | 20 db/octave |
| Pad Fast Fourrier Transform | 500 samples |
| Amplitude scaling | 2D |
| Derivative filter | 2D |

## Appendix C

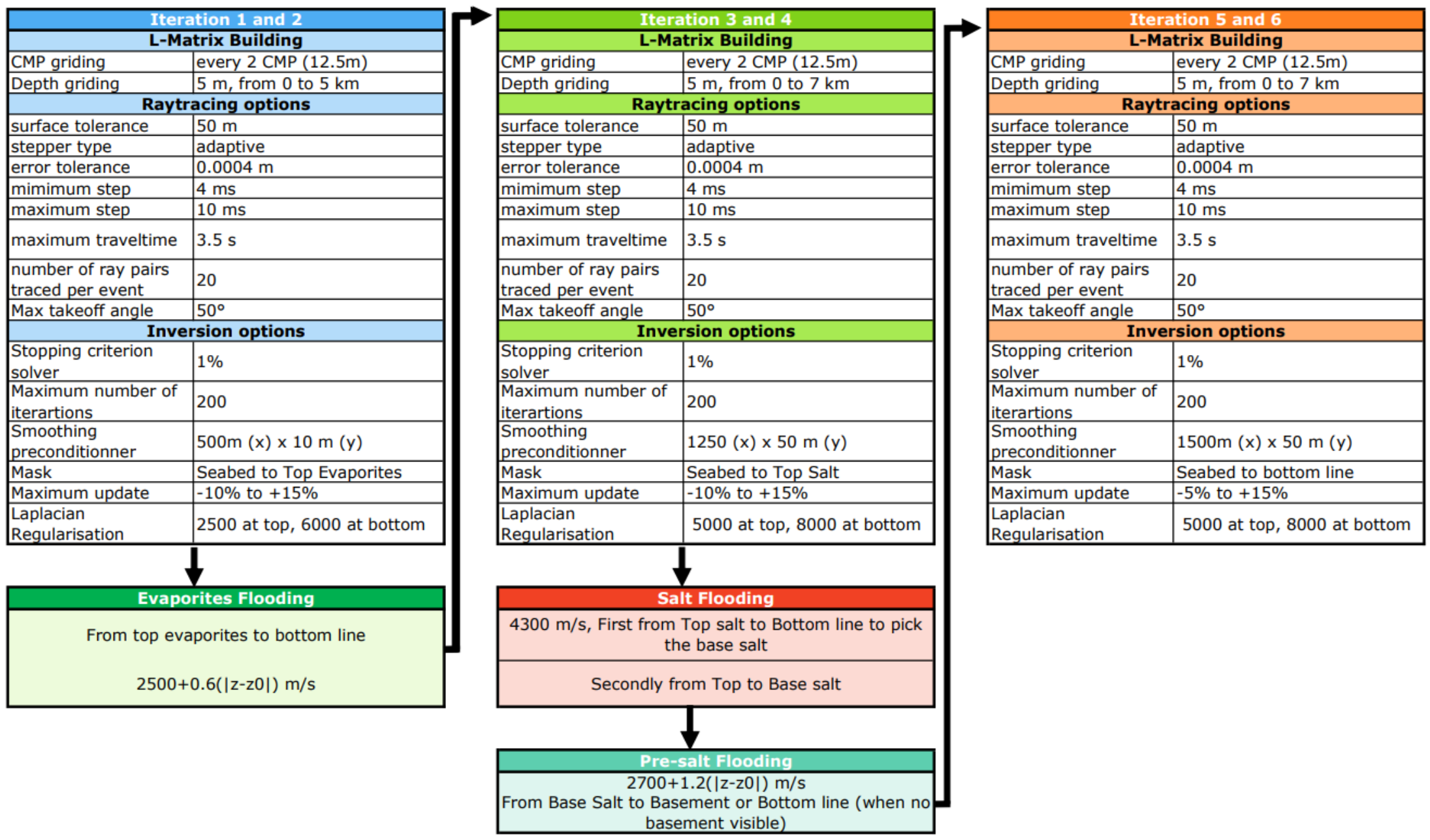

**Figure C1 Depth imaging flow and list of the parameters used for the different tomographic inversion and velocity updates during the velocity model building**

### 9. Author contributions

SB performed the data processing, drafted the manuscript, and created the figures. JF contributed to the data processing and the final versions of the figures and manuscript. AL, ADB, and AC contributed to the manuscript revision and the project supervision.

### 10. Competing interests

The authors declare that they have no conflict of interest.

### 11. Acknowledgements

This research is carried out under the SALTGIANT ETN, a European project funded by the European Union's Horizon 2020 research and innovation programme under the Marie Skłodowska-Curie grant agreement n° 765256.
The SALTFLU seismic profiles were acquired within EUROFLEETS, call for ship-time 'Ocean' 2010, project 'Salt deformation and sub-salt fluid circulation in the Algero-Balearic abyssal plain – SALTFLU'.
The seismic data have been processed using REVEAL© processing software generously provided to the University of Trieste by Shearwater Geoservices. Data were interpreted using Petrel©, generously provided to the University of Trieste by Schlumberger.

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
