# Peer review of "Figure 1 Study area offshore Balearic Islands, Spain. Seismic lines acquired by the National Institute of Oceanography and Applied Geophysics (OGS) are superimposed on a bathymetric map and on the relief map of the Mediterranean with DSDP and ODP drillsites (in red the ones that sampled MSC evaporit"

_Earth System Science Data, 2022_

## Referee Comment (RC1)

**Review for „Reprocessed 2-D airgun seismic reflection data SALTFLU (salt deformation and sub-salt fluid circulation in the Algero-Balearic abyssal plain) in the Balearic promontory and the Algerian basin"**

**General Comments**

This paper provides an interesting and unique collection of reprocessed multichannel seismic data from the Algerian basin and the Balearic promontory. The authors present a sophisticated processing flow enabling the challenging task of imaging the base of the Messinian. Through a broadband processing strategy, multiple attenuation, and iterative migration in the time and depth domain, the authors produce seismic images allowing an interpretation of the Algerian basin and the Balearic promontory from the Miocene up to the Plio-Quarternary strata. The study provides several interpretations of the intra-Messinian deposits, the salt tectonic system, fluid migration structures, and volcanic structures.

In my opinion, the data set and the presented processing scheme will be a valuable contribution to the community, especially since comparable data cannot be acquired in the near future due to the ongoing exploration efforts in the area. However, I recommend that the manuscript should be revised since I came across several major issues, which I list below before providing more detailed comments.

1. Structure of the manuscript

   In the current version of the manuscript, the section 'Results' seems to be a mixture of a summary of the processing flow and the discussion. Technically, the results are already included in Chapter 3 'Methods', presenting the results of the processing scheme. I suggest focusing Chapter 4 on the comparison of the reprocessed data and the vintage data since this is key to this study, which brings me to my next point:

2. Reprocessed vs. vintage data

   In my opinion, a key aspect of the manuscript is that the new processing scheme provides new geological insights compared to the vintage data. However, this is hardly addressed in the text and it is not illustrated well enough in the figures, in my opinion. While Figure 6 shows two examples of a comparison of selected reprocessed and vintage sections, I think this important aspect of the manuscript could be strengthened by the inclusion of more examples, especially by showing more detailed figures (e.g., enlargements of the sub-salt). In addition, in my opinion, the comparison of seismic line SF09 in Figure 6 seems a bit unfair since the amplitudes of the vintage section seem to be weaker in general, which makes a comparison difficult.  I am aware that the processing scheme improves the balance of the amplitudes but in order to enable a visual comparison for the reader, I recommend scaling the data similarly (e.g. an RMS scale or a Mean scale).
   In addition, I recommend that the authors also upload the vintage versions of the profiles to the repository so that the reader can evaluate the improvements in the presented processing scheme.

3. Data Completeness

   I realized that there are several issues with the data uploaded to the repository at zenodo.
   - For SF09, the PSTM version and the velocity model are missing.
   - In SF12, the data quality of the last third of the profile is very bad. Something clearly went wrong here, see Figure below:

[Figure]

*Figure 1: Profile SF12 (PSTM). The yellow line marks an area where the quality of the seismic section strongly deteriorates.*

4. Figures

   There are several figures, where it is very difficult for the reader to properly follow the reasoning of the authors. On the one hand, this is due to the limited resolution of some of the figures (e.g. Figs. 3, 4). On the other hand, several images are too small to properly assess the differences before and after processing steps (e.g. Fig. 3) or to follow the interpretation (e.g. Fig. 10). It would help if these figures were made larger and displayed in better resolution.

   Also, I think the manuscript would benefit if the authors included more references to figures in the text and were more specific about where exactly the reader should look. Also, in some instances in the text, the figures are not referenced at all (e.g., Fig. 4e, Figure 7a-c). Further, I recommend being consistent with the figure captions, i.e. to always explain what the subfigures (a), (b), etc. show. This is not the case in Figures 1, 4, 8, 10, and 12. I think it would also be helpful if the authors provide some sort of horizontal scale along the seismic section so that they can better direct the reader's eye to specific areas of the profiles.

5. Data Quality

   In the Review Criteria of ESSD, it says: "The data must be presented readily and accessible for inspection and analysis to make the reviewer's task possible. [...] The reviewer will then apply his or her expert knowledge and operational experience in the specific field to perform tests (e.g. statistical tests) and cast judgement on whether the claimed findings and its factors – individually and as a whole – are plausible and do not contain detectable faults."

   If I understand this correctly, the raw seismic data and navigation data should also be uploaded in order to make the workflow reproducible. In addition, as mentioned above, it

would be great if the vintage version of the data would be available to properly assess the improvement of your presented processing workflow.

6. Presentation Quality

In the Review Criteria of ESSD it says: "The authors should point to suitable software or services for simple visualization and analysis, keeping in mind that neither the reviewer nor the casual "reader" will install or pay for it".
I think this is something you should address in the manuscript. While this is of course challenging when seismic data are involved, I would recommend the following:
- You can point out free software packages that allow the reader to plot the seismic sections (e.g. Seismic Unix, SeisSee, etc.)
- You can provide high-quality images (e.g. PDFs) of all profiles so the reader can immediately assess the profiles and get an easy impression of the dataset.

7. Geological Intepretation

In general, I had major problems following the interpretation of the seismic profiles. I suggest the following modifications:

- The relation between reflectors (R1-R5) and units (UU, UU1, UU2, intra-UU) should be clearly defined at the beginning of Section 5.3 or 5.3.2. The text jumps between both nomenclatures and this is very confusing. Also, the labels in the Figures are not consistent (see comment on Figures 10 and 11 below). Please make sure that the colors used for horizons are consistent.
- Provide a horizontal scale along the profiles so you can specifically refer to the profile kilometer where the geological feature of interest is located. In many cases, I had to search for a specific area for a long time, which is very distracting from the line of reasoning in the text.
- In Figure 10 I strongly recommend placing the blow-ups in a separate panel below and to plot them larger. Otherwise, it is impossible to evaluate and understand the interpretation.
- Please make sure that all features indicated in the Figures are mentioned in the text, e.g. "LAV" in Figures 8 and 11, "gas pocket" in Figure 10," "salt lenses" in Fig. 12a, "intrusive sills" in Fig. 12c are not mentioned in the text at all.

8. Quantification of uncertainties

The authors mention that there are uncertainties regarding the final velocity model due to the limited offset. However, I strongly recommend addressing these uncertainties better in the manuscript and to discuss the resulting uncertainties in the depth of seismic reflectors, e.g. the base of the evaporites.

9. Referencing sections

The authors consistently refer to "Section 2", where it should be "Section 3". E.g. in Lines 237 or 244, the authors refer to "Section 2.4.1" or "Section 2.4.2", which do not exist in the manuscript.

10. Table 1 is missing in the manuscript.

**Figures**

Figure 1:

- Map in (a) misses proper coordinates and information regarding the coordinate system.
- For the overview map in (b), I would recommend removing the coordinate lines and adding ticks to the axis instead.
- The Figure caption does not explain (a) and (b).
- The Fonts are very small in (b). The numbers of the drill sites are very difficult to see.

Figure 3:

I think that the quality of these plots should be improved. In order for the reader to properly evaluate the improvements in the processing, please provide larger Figures in (b) and (c). In addition, the frequency plots are very difficult to see and I would recommend providing larger versions of the plots. In addition, I strongly recommend deleting the horizontal and vertical lines since they are not consistent (some are missing?) and are not really necessary.

Figure 4:

- Half of the figure caption is missing.
- I like the panels in (a) a lot and the improvement of the deghosting is impressive!
- In (b) the inconsistent vertical dashed lines give the impression of a logarithmic scale. Please delete these annotations, they are very distracting.

Figure 5:

- While I really like the larger representation of the figure that allows the reader to properly assess the improvements of the demuliple, the legend is incomplete. The meaning of the orange line (base of evaporites, I assume) is not indicated in the legend.

Figure 6:

- While you state that the amplitudes are balanced batter in the reprocessed sections, it seems that the amplitudes of the vintage data are much weaker in general. For comparison, it would be helpful to apply an RMS or mean scaling to better illustrate the differences between both versions.
- As mentioned above, I think you could really improve the story of the manuscript by providing more detailed illustrations showing the improvements of the processing flow, especially regarding the imaging of the sub-salt strata.

Figure 8:

- This is a very nice figure! The resolution is perfect here.
- The explanation for the abbreviation LAV is missing.
- Please add (a) and (b) for the upper and lower panel.

Figure 9:

- In the caption, it says: "(a) common reflection-point gathers with interval velocity overlay". However, there is no interval velocity overlay in (a).

Figure 10:

- The colors of the horizons are not consistent. The base of the salt/evaporites has a different color in (a) and (c) compared to (d) and the legend.
- Please make sure the Figure caption is consistent. The explanation for panel (c) is missing.
- It is really difficult to link the text with the Figure. I would recommend placing the enlargements in a separate panel below and making them larger. In addition, it would greatly help to place arrows in the figures and a horizontal scale so that it is easier to follow your reasoning in the text. E.g., where are exactly are weakly reflective diapir contacts exactly? Where exactly are the fluid indicators? An experienced seismic interpreter can see this, but this might not be clear to other readers.

Figure 11:

This Figure does not allow the reader to understand the interpretation that is outlined in the text. It is not possible to identify the different reflectors (R5, R4, R3) since the markings are unclear.
In the caption, it says that reflector R5 is flattened, so I assume that this is the green dashed horizon, which is perfectly flat. However, this is not indicated in the legend. Then again, I only see one more marked horizon in black – is this R4 or R3? I don't see another marked reflector as there should be according to the legend. In the legend, it says R4 is the "top evaporites top surface", which makes no sense to me, especially since the caption says R5 is the top of the Messinian evaporitic sequence.
I don't want to be too picky or too harsh here, but for me, this makes it impossible to follow your reasoning in the text.

Figure 12:

- (B): The interpretation of magma conduits is speculative and not necessary. While the interpretation of the irregular high-amplitude reflections as volcanic terrain is convincing, the seismic data do not allow to say anything about the internal structure of these edifices such as conduits.
- (C): It is not possible to identify the collapse structures you indicate in this figure. The amplitudes of the unit containing the evaporites are too strong to identify any internal structures here.
- (C): It is impossible for the reader to properly identify what is interpreted here as mass-transport deposits since it the figure is way too small.
- (C): The interpretation of intrusive sills is very speculative and not addressed at all in the text.

Appendix B/C:

- Inconsistent use of "meters" or "m"
- Missing unit for the aperture?
- Please make sure that there is a space between the numbers and the units. This is not consistent in the tables.

**Text**

Line 12, 16:

- Your data was acquired with a streamer of 3 km length, which you term "short-offset" here. I understand that the offset-to-target ratio is not great, especially for the aim of imaging the sub-salt strata, but I disagree with this terminology. In the context of academic seismic surveying, where often only single-channel seismic data or streamers with lengths much shorter than 1000 m are available, the used streamer length of 3000 m is a lot. Also, a streamer length of 3 km is more or less the limit of what can be handled from research vessels.

Line 48:

- Table 1 is missing!

Line 55:

- It would help to include a figure here that highlights the weaknesses and processing challenges of the vintage data.

Lines 76-80:

- When you mention the occurrence of the 'bubble pulse', I think you should mention that for the survey, G.I. guns were used that are specifically designed to suppress the bubble signal. While this suppression is not 100 %, I think it is necessary to at least mention this here.

Line 110:

- I assume that you mean 'common image gathers' or 'common reflection-point gathers' instead of 'coming midpoint gathers' here?

Line 120:

- Otherwise, I really like the Introduction. It nicely sets the motivation of the study and gives a good overview.

Line 122:

- In Figure 1, you capitalize "Formentera Basin" and "Algerian Basin" but in the text, you use the lowercase version "Formentera basin" and "Algerian basin". This should be consistent.
- The abbreviation EBE is only used once. I would recommend deleting it.

Line 132:

- The reference (Lofi, 2018) is not in the reference list. Should this be (Lofi, 2011)?

Line 148:

- I think you should refer to Figure 1b when mentioning ODP Site 975, which by the way is difficult to see in the small inlet.

Line 150:

- I was wondering, did you apply a topmute somewhere in your processing routine, e.g. after migration? The data looks like there has been some kind of topmute applied but this is not indicated in Figure 2 or mentioned in the text.

Line 169:

- How did you estimate 1511 m/s as the water velocity? I understand that ~1500 m/s makes sense, but 1511 m/s gives the idea of a precision that needs explanation.

Line 211-219:

- This paragraph should refer more thoroughly to Figure 4. Only Figures 4a and 4d are mentioned here, while the rest of Figure 4 is not mentioned at all. I think it is necessary to better match the text and the figures, otherwise, the reader is lost.

Line 279:

- In my opinion, it would make sense to place Figure 7 or another figure illustrating the velocity building scheme somewhere in this section.

Line 281:

- Here, you are using 1525 m/s as water velocity. Why not use 1511 m/s? Or why did you not use 1525 m/s before?

Line 288:

- What tomography code did you use? Is this included in the REVEAL processing software?

Line 302:

- It would be great to shortly discuss the derived value for the salt velocity. Is this a reasonable value compared to other studies?

Line 355:

- Please indicate that you used the λ/4 criterion here.

Lines 389-390:

- How did you calculate the notch frequencies? I recommend shortly explaining this.

Lines 399-400:

- Please indicate whether birds were used during the acquisition of the seismic data for depth control

Line 446:

- Please explain, why you "believe" they represent geology. I agree, especially  since this is actually quite clear in Figure 8b due to the hummocky, irregular appearance of the internal reflector compared to R2.

Line 498:

- CMP gathers – This should probably be common reflection-point gathers.

Lines 449-450:

- The reference to Figure 9 seems misleading to me since Figure 9 does not show the over-migration of R2.

Line 459:

- You could mention the offset-target ratio here.

Line 460:

- What are the uncertainties in the velocities? Can you quantify this?

Line 501:

- This is very difficult to see. I think providing larger figures would definitely help.

Line 506:

- The low amplitudes and push-downs are not indicated in the figure.

Line 507:

- There is no Figure 8d

Line 509:

- Please refer to which reflector the reader should look here and mark where reflectors are pierced.

Lines 506-530:

- I have the feeling that this paragraph needs to be reformulated to be understandable by an inexperienced reader. I think it is necessary to guide the eye of the reader better and to state what features the reader should look for since this is not always clear in the figure. (e.g. by referring to arrows).

Line 530:

- Figure 10a indicates gas pockets. However, this is not included in the text.

Line 537 -576:

- It is extremely difficult to follow the line of reasoning here. Please make sure to refer to your figures more specifically to guide the reader's eye.
- In addition, it is quite confusing that the text switches between the Units (UU1, UU2, etc.) and the reflectors (R1-R5). I think it would be very helpful to clearly define how the units and reflectors relate to each other at the beginning of this section.

Line 545:

- As far as I can see, those faults lie above reflector R1 and not below.

Line 548:

- Please indicate the reflector truncations in the figure, it is really difficult to understand where the reader should look here.

Line 563-565:

- Since it is unclear where R4 is in this Figure, it is not possible to follow the reasoning here.

Line 568:

- It would be helpful to mark the Balearic margin in Figure 11.

Line 578:

- The abbreviation LU is not defined anywhere.

Line 583:

- The abbreviation FBI is only used twice in the text. I would recommend removing it.

Figure 607:

- The collapse structures are highlighted in Figure 12c. Therefore I strongly recommend referring to Figure 12c here to help the reader follow the line of reasoning.

Line 613:

- Vertical dimming is not shown in the figure or at least it is not clear here.

Line 615:

- In the figure, you indicate salt lenses, but don't touch on it in the text. This should be consistent.

Line 649:

- I disagree that the reprocessed data reveal these features. Rather than that, I would totally agree that reprocessing helps to image these structures in greater detail and more precision. Therefore, I recommend rephrasing this or providing examples that justify this claim.

**Technical corrections**

Line 103: window based → window-based

Line 164, 313, 320, …: Sometimes you are not using a space between the numbers and the unit. Please check the entire manuscript again, this must be consistently avoided.

Line 184: Double space: "domain __ prediction"

Line 468: du to → due to

Line 469: limit → limits

Line 526: The → Since the

Line 594: that → than

Line 184: Double space? "similar __ to"

Line 612: Remove space after reference → (Bertoni and Cartwright)_.

Line 625: Missing brackets at the reference: Infante-Paez and Marfurt, 2017.
* * *
I am at your disposal for any questions.

Kind regards,

Jonas Preine, University of Hamburg

Email: jonas.preine@uni-hamburg.de

---

## Referee Comment (RC2)

The manuscript is entitled **Reprocessed 2-d Airgun seismic reflection data Saltflu (salt deformation and sub-salt fluid circulation in the Algero-Balearic abyssal plain) in the Balearic promontory and the Algerian basin**.

Main objective of the present manuscript is to better imaging of the salt structures (particularly base salt) which was interpreted as flat lying layer cake in the previous processing and interpretations. Several challenges face the authors including; velocity variation, eliminate multiples and improve signal to noise ratio at depth. The authors tried to overcome these challenges through introducing a new technique for processing of the 2D multi channel seismic reflection data set through the following stages; broadband processing; multi-domain denoising and demultiple and construction of the geologically guided velocity model utilizing iterative pre-stack migration and travel time tomography. The manuscript is well written but need some modification in order to be suitable for publication. The present manuscript requires corrections.

1) The Geologic setting section in this manuscript requires more elaboration about the stratigraphic framework and tectonic setting of the study area.

2) The manuscript provides new technique for the processing legacy 2D seismic data and new way to build detailed velocity model (show enhanced salt top, salt base and pre-salt reflectors). New features as DHI and mud volcanoes have been observed as result of such processing, the reviewer recommend (**if available**) to reapply the same processing technique in area with same complex conditions (water depth, salt deformation,…… etc) in area with at least 3 wells containing velocity data (preferred VSF data) in order to correlate the well velocities to seismic velocities (specially with presence of high velocity salt thick layer) and better understanding the acquired data as non-zero phased data (which will affect the processing results signifantlly).

3) Severe smoothing of the velocity model and approximation of the navigation data puts the output seismic sections are much uncertain and need to be taken as regional context.

4) Compare the mud volcanoes with some regional surface exposures of the mud volcanoes as Nirano mud volcano in Italy, it would be great if you construct map showing hypothetical distribution of DHI in the study area.

5) Construct a table that summarizes the seismic facies showing the new subdivisions of the Messinian Upper and Lower Units (obtained from the newly created seismic profiles) because it is unclearly presented and makes controversy to the reader.

6) Construct a table that summarizes the correlation of the seismic facies obtained from Mallorca Basin and the seismic facies of the Belaric Promontory because it is unclearly presented and makes controversy to the reader.

7) Add (Feng and Reshef, 2016) and (Jackson and Hudec, 2017) to the references list.

**Figures:**

1) **Figure No. 1**
   - Enlarge Figure.1b, highlight the location of ODP site 975 and add scale bar to the map.
   - Add coordinates to the figure.1a
   - Add location of Alger-1 well.
   - Add location of the Mallorca Basin.
   - Add detailed stratigraphic column for the study area showing stratigrahic aspects and main tectonic events.
   - Add schematic geoseismic line showing structural configuration of the study area, soothe reader would be able to comprehend the complexity the velocity model.

2) **Figure No.9**
   Add directions NW-SE along the line SF06.

3) **Figure No.10**
   Add directions NW-SE along the lines SF03, SF08 and SF09.

---

## Referee Comment (RC3)

Review of "Reprocessed 2-D airgun seismic reflection data SALTFLU (salt deformation and sub-salt fluid circulation in the Algero-Balearic abyssal plain) in the Balearic promontory and the Algerian basin" by Blondel *et alii*

**General comments**

The main focus of the paper is the method applied to reprocess legacy seismic data. Most of the text is dedicated to this issue. The title is therefore somehow misleading. I think that changing "Reprocessed" into "Reprocessing" (as it is stated at the beginning of the paragraph "Conclusions") would be appropriate.

The method adopted combines several processes to different phases of seismic data elaboration in order to improve the quality of the final output. However, it is not always clear how each step has been selected by the authors and whether it corresponds to any automated protocol of the REVEAL software used.

Geological insight obtained by reprocessing the original dataset provides evidence of the enhancement of data imaging achieved through the method described which could be very useful in reprocessing other sets of legacy seismic data.

**Specific comments**

The paper would benefit by a more thorough comparison between legacy data and the present elaboration. Each paragraph of the "Methods" section would be more easily understood if complemented by a figure showing legacy data facing current results in order to point out differences and improvements (as it was done for figure 6).

I suggest to clearly separate the discusssion concerning the reprocessing (which is a real discussion of results) from the geological interpretation which includes evidences derived from the literature and is used to support the improvements achieved through the applicaton of the present method. Paragraph 5.3 might better constitute a separate paragraph 6 renamed "Geological implications" to be put before 7 "Data availability" and then 8 "Conclusions". This last paragraph is appropriate and corresponds to what the paper describes.

The main geological considerations concerning salt units are actually the subject of a paper already published (Blondel et alii, 2022) in which the present paper was cited as in prep. The only additional geological information seems to concern potential mud volcanoes.

Altogether, the authors should pay more attention to details and cross references (e.g. fig.4 is missing part of the caption), as also reported below in the Technical corrections.

**Technical corrections**

Line 48: table 1 is not present in the paper

Line 118: it is better to rephrase: "These reprocessed images allow to highlight …", since this is not thoroughly described in this paper.

Line 136: it would be better to represent the location of borehole Alger 1 in the figure

Line 223: delete depth

Line 237: reference should probably be to section 3.4.1

Line 244: reference should probably be to section 3.4.2

Line 244: Please, explain the abbreviation CDP as it has been done in other cases

Line 265: reference should probably be to section 3.3

Line 277: reference should probably be to section 3.4.1

Line 277: reference should probably be to section 3.4.1

Line 311: reference should probably be to section 3.3

Line 317: reference should probably be to section 3.2

Line 363: reference should probably be to section 5.3.2

Line 468: change du into due

Line 503: delete are

Line 508: change supports into support

Line 593: add as before that, i.e. as that

Line 625: citation should be put in brackets, i.e. (Infante-Paez and Marfurt, 2017)

Line 629: add basin after Formentera